# Interactome disassembly during apoptosis occurs independent of caspase cleavage

Nichollas E Scott[1,2], Lindsay D Rogers[3,4], Anna Prudova[1,2], Nat F Brown[1,2], Nikolaus Fortelny[1,2,4], Christopher M Overall[2,3,4] & Leonard J Foster[1,2,*] iD

## Abstract

Protein–protein interaction networks (interactomes) define the functionality of all biological systems. In apoptosis, proteolysis by caspases is thought to initiate disassembly of protein complexes and cell death. Here we used a quantitative proteomics approach, protein correlation profiling (PCP), to explore changes in cytoplasmic and mitochondrial interactomes in response to apoptosis initiation as a function of caspase activity. We measured the response to initiation of Fas-mediated apoptosis in 17,991 interactions among 2,779 proteins, comprising the largest dynamic interactome to date. The majority of interactions were unaffected early in apoptosis, but multiple complexes containing known caspase targets were disassembled. Nonetheless, proteome-wide analysis of proteolytic processing by terminal amine isotopic labeling of substrates (TAILS) revealed little correlation between proteolytic and interactome changes. Our findings show that, in apoptosis, significant interactome alterations occur before and independently of caspase activity. Thus, apoptosis initiation includes a tight program of interactome rearrangement, leading to disassembly of relatively few, select complexes. These early interactome alterations occur independently of cleavage of these protein by caspases.

**Keywords** caspase; protein complexes; protein correlation profiling; SILAC
**Subject Categories** Autophagy & Cell Death; Genome-Scale & Integrative Biology; Post-translational Modifications, Proteolysis & Proteomics
**Mol Syst Biol. (2017) 13: 906**

## Introduction

The association of proteins into functional units is a quintessential feature of life, with 150,000–650,000 discrete protein–protein interactions predicted within the human proteome (Hart *et al*, 2006; Stumpf *et al*, 2008). Functional flexibility and specificity of complexes occur by the exchange of subunits (Russell *et al*, 1999; Wilson *et al*, 2008), with fine-tuning of protein abundances

modulated through the stabilizing/destabilizing effects of interactions (Daley, 2008; Mueller *et al*, 2015). The recognition that proteomes are highly interconnected networks, or interactomes, has changed biology's view of cause and effect, simultaneously highlighting the importance of systems biology and necessitating its use in deciphering complex biological phenomena effects (Barabasi & Oltvai, 2004; Barabasi *et al*, 2011). This concept of connectivity modulating systems explains how functionally dissimilar systems, such as tissues, can have largely similar compositions (Geiger *et al*, 2013; Kim *et al*, 2014; Wilhelm *et al*, 2014). Through viewing systems as networks, it has become increasingly clear that it is more than just the presence of a specific allele (e.g., a disease-causing mutation) that leads to phenotypes; rather, it is the effect of the allele on the network (Vidal *et al*, 2011; Menche *et al*, 2015; Sahni *et al*, 2015). Indeed, mutations in different members of a protein complex can lead to the same phenotype with these highly connected groupings known as disease modules (Menche *et al*, 2015). The very existence of such modules suggests that a network-level view of the proteome is more meaningful for understanding biological states and disease than a detailed picture of specific components (Bandyopadhyay *et al*, 2010; Barabasi *et al*, 2011; Califano, 2011).

Apoptosis is a classic example of a complex phenotype mediated by dynamic protein–protein interactions (PPIs) (Tait & Green, 2010; Crawford *et al*, 2012; Kaufmann *et al*, 2012). Apoptosis can be initiated by multiple pathways (Wajant, 2002; Elmore, 2007; Riedl & Salvesen, 2007), with the surface receptor CD95 (APO-1/Fas) playing an essential role in immune homeostasis (Siegel *et al*, 2000). Initiation of Fas-mediated apoptosis starts with aggregation of Fas receptor (Kischkel *et al*, 1995) and proceeds via two distinct downstream pathways that differ in the role of mitochondria (Scaffidi *et al*, 1998), but ultimately culminate in caspase activation. To counterbalance apoptosis initiation, multiple systems limit the propagation of caspase activation through rapid proteasome-mediated turnover (such as in the case of activated caspases 3, 6, and 7; Suzuki *et al*, 2001; Gray *et al*, 2010), or by direct inhibition, for example, by binding of X-linked inhibitor of apoptosis protein (XIAP) to caspase-7 (Huang *et al*, 2001). If left unchecked, initiator caspases amplify the activation of the executioner caspases 3 and 7,

1  Michael Smith Laboratories, University of British Columbia, Vancouver, BC, Canada
2  Department of Biochemistry and Molecular Biology, University of British Columbia, Vancouver, BC, Canada
3  Department of Oral Biological and Medical Sciences, University of British Columbia, Vancouver, BC, Canada
4  Centre for Blood Research, University of British Columbia, Vancouver, BC, Canada
    *Corresponding author. Tel: +1 604 822 8311; E-mail: foster@chibi.ubc.ca

committing the cell to death through the cleavage of multiple substrates (Crawford & Wells, 2011; Crawford *et al*, 2012; McIlwain *et al*, 2013). An open question remains: Do only a select few substrates need to be cut in order to pass checkpoints in programed cell death or is it a "death from a thousand cuts" (Martin & Green, 1995; Chowdhury & Lieberman, 2008) that overwhelms system robustness? Interestingly, caspases target protein complexes more frequently than stand-alone proteins and sometimes cut multiple members of the same complex (Mahrus *et al*, 2008; Stoehr *et al*, 2013), suggesting that cleavages may cause complex inactivation or disassembly. This currently uncharacterized yet potentially critical connection between proteolysis and interactome alterations in apoptosis is vital for understanding and treatment of diseases such as autoimmunity and cancer.

Over the last decade, our knowledge of PPI networks has expanded (Gavin *et al*, 2002, 2006; Guruharsha *et al*, 2011; Babu *et al*, 2012; Havugimana *et al*, 2012; Rolland *et al*, 2014), but our understanding of contextual interactomes is far from complete. Large mapping efforts (Malovannaya *et al*, 2011; Rolland *et al*, 2014; Huttlin *et al*, 2015) provide an invaluable resource for the scientific community, yet, by necessity, are limited in their scope due to poor scalability, which limits interactome characterization to a select few cell types under artificial expression conditions (Ewing *et al*, 2007; Huttlin *et al*, 2015), typically using only one protein isoform (Team MGCP *et al*, 2009; Yang *et al*, 2011). Thus, current networks remain of limited use in interpreting biological phenomena, as the observed interactions may not reflect the interactome landscapes within other systems or in disease. Alternative approaches are needed for mapping of native, endogenous interactomes under multiple conditions. Co-migration approaches, such as protein correlation profiling (PCP), represent one such solution (Kristensen & Foster, 2013; Larance & Lamond, 2015).

Protein correlation profiling and related co-migration-based approaches for protein interaction analysis (Havugimana *et al*, 2012; Kirkwood *et al*, 2013; Borner *et al*, 2014) enable the assignment of interactions based on the similarity of migration of proteins across a non-denaturing separation gradient, with the underlying assumption that the association of proteins is responsible for the observed similarity (Oliver, 2000). Using this principle, we and others have previously shown that thousands of interactions could be assessed within a single experiment (Havugimana *et al*, 2012; Kristensen *et al*, 2012; Scott *et al*, 2015; Wan *et al*, 2015)—with the incorporation of stable isotope labeling by amino acids in cell culture (SILAC)-based multiplexing further enabling the simultaneous comparison of multiple states (Ong *et al*, 2002; Kristensen *et al*, 2012). These co-migration-based technologies now make it possible to characterize interactomes in response to stimuli on a scale previously inaccessible, facilitating true comparative interactome studies. The key potential of this technology is the ability to explore disease states where protein interaction alterations are suggested yet the scale and proteins subjected to alteration are unknown.

Here we combine two quantitative proteomics techniques, PCP and TAILS, to characterize apoptosis-induced changes in protein interactions as a function of caspase activity (Fig EV1). Using Jurkat T cells as a model, we find that most mitochondrial membrane and cytosolic interactions are unaffected by Fas-mediated apoptosis at 4 h, yet discrete alterations are detected in both compartments,

suggesting their mechanistic importance in initiating cell death. In parallel, we show that although apoptosis-dependent proteolytic events are evident, dramatic alterations in the interactions of previously described and novel caspase targets appear to be independent of their cleavage. Rather, our data suggest that the rearrangement of the interactome during apoptosis initiation is the harbinger of proteolysis, not the result of it.

# Results

## Adaptation of PCP-SILAC to study dynamic mitochondrial/membrane interactomes

Studies of interactome-wide changes are rare but they are non-existent for organelle or membrane interactomes. As PCP-SILAC enables the measurement of cytosolic interactome responses (Kristensen *et al*, 2012), we reasoned that using a membrane-compatible separation method should allow the measurement of organelle/membrane interactome dynamics. SEC provides a robust workflow for the separation of cytoplasmic complexes; however, it is not compatible with membrane complexes as they are extremely sensitive to separation conditions (Drew *et al*, 2008; Babu *et al*, 2012). Thus, to analyze membrane protein complexes, we utilized PCP-SILAC and BN-PAGE, a separation approach known to be broadly applicable to membrane complexes (Wittig *et al*, 2006) (BN-PCP-SILAC, Appendix Fig S1). Using this approach, we identified a total of 4,363 protein groups from a mitochondrial membrane preparation solubilized with digitonin (Table EV1); within this dataset, each protein group quantified in an average of 19 BN fractions (39%; Appendix Fig S2A and Table EV2) and > 70% of previously identified proteins from in-depth BN study of mitochondria preparations (Heide *et al*, 2012) observed (Appendix Fig S2B). Initially, we used the medium and heavy SILAC channels to compare two technical replicates for reproducibility (Appendix Fig S1). Applying our bioinformatics strategy (Kristensen *et al*, 2012; Scott *et al*, 2015) where Gaussian curves are fitted to each protein chromatogram, a total of 4,632 curve Gaussian features from 2,384 protein groups were determined (Tables EV3 and EV4) across the two experimental isotopologue channels (i.e., technical replicates differentially labeled medium and heavy) (Fig 1A). Reproducibility of quantitation between isotopologue channels was very high (98%, Appendix Fig S2C), proving that this approach can accurately quantify interactome changes.

We constructed a mitochondrial membrane interactome map by considering the Euclidean distance and fitted curve of all protein profiles to all other profiles, as previously described (Kristensen *et al*, 2012; Scott *et al*, 2015). This approach, unlike techniques which assess only direct interactions such as yeast two-hybrid, enables both direct and indirect interactions between proteins found within the same supramolecular to be identified. As 30% of observed proteins in our dataset are also in the Comprehensive Resource of Mammalian protein complexes (CORUM) database (Table EV5) (Ruepp *et al*, 2010), this allowed similarity cut-off based on Euclidean distances and fitted curves to be determined using known, gold-standard interactions pairs. Using this approach, we identified a total of 6,436 unique interactions at a global precision (i.e., the combined precision of all interaction across all

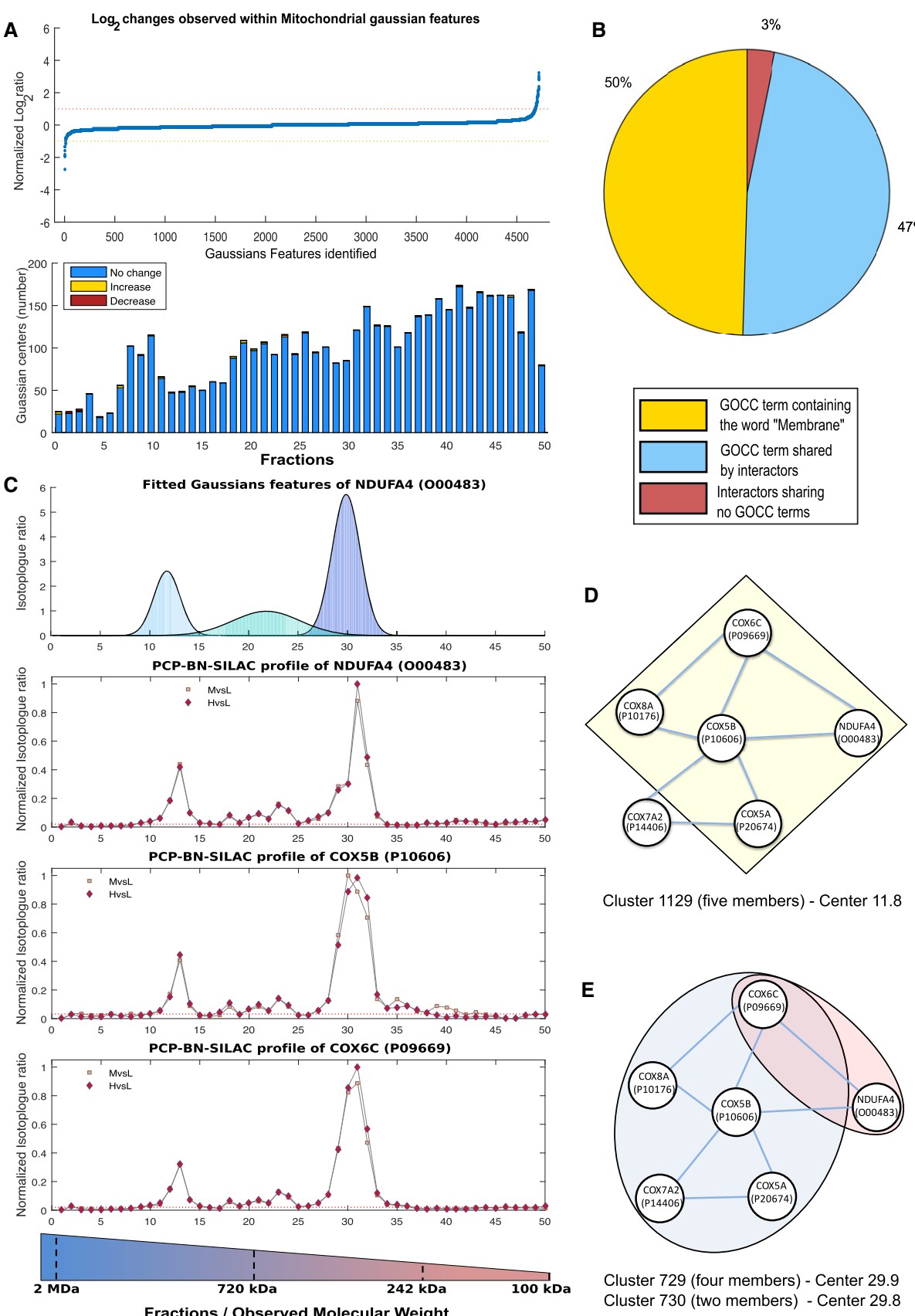

**Figure 1.**

**Figure 1.  Validation of the BN-PCP-SILAC approach.**

A    Observed changes in the interactome of Jurkat-derived mitochondrial/organelle samples prepared in parallel, a total of 4,632 Gaussian features were mapped across 50 fractions with few alterations within the interactome observed.

B    Pie chart of observed GO terms within detected interactions, ~97% of interactions corresponded to membrane-associated proteins.

C    Protein profiles of Complex IV members O00483, P10606, and P09669 all display identical elution profiles and stoichiometry supporting the assignment of the interaction. The 50 fractions of the BN-PAGE separation span a mass range of 2 MDa to 100 kDa, which is visualized within the provided mass gradient.

D, E  Markov clusters resulting in the correct assignment of O00483 as part of Complex IV. Within the identified clusters, multiple clusters containing O00483 were identified including a five-member cluster centered at fraction 11.8 (D) and a binary cluster centered at fraction 29.8. By further grouping clusters based on co-elution and shared membership, cluster 729 and 730 (E) are observed to generate the identical cluster observed at fraction 11.8.

isotopologue channels) of 66% with a FPR of 0.1% (Table EV6A). Consistent with the power law behavior of biological systems, the resulting interaction network demonstrated a strong linear trend typical of scale-free networks (Barabasi & Albert, 1999; Fig EV2A). As expected for a membranous sample, 97% (6,232 out of 6,436) of interactions within the detected network included at least one partner associated with a membrane cellular compartment Gene Ontology (GO) term (Fig 1B and Table EV7). Further, 50% (3,188 out of 6,436) of interactions between protein groups were associated with the same membrane-related GO cellular compartments term (Fig 1B and Table EV7), consistent with the determination of authentic membrane protein interactions.

Interestingly, even at reduced precision levels, the observed FPR was extremely low (0.25 and 0.5% for local precision thresholds of 60 and 50%, respectively, Tables EV6B and C, and EV8), indicating that our approach yields very high confidence interactomes, further supported by the precision-recall metrics (Fig EV2B). Further, we used Markov clustering (Enright *et al*, 2002; Guruharsha *et al*, 2011; Babu *et al*, 2012) to assemble binary interactions into higher order associations, or complexes, resulting in 2,060 unique clusters across the BN separation gradient (Appendix Fig S3A, 70% local precision). Consistent with our approach showing favorable characteristics for the identification of multi protein complexes, the use of Markov clustering resulted in a marked increase in the precision (83% final precision) of the determined membrane interactome (Appendix Fig S3B and C, and Tables EV9 and EV10).

BN appears to be an excellent fractionation strategy for PCP-SILAC experiments, as illustrated by the high resolution of the mito-chondria-associated protein cytochrome c oxidase subunit NDUFA4 (O00483) interactions. Originally thought to associate with Complex I (Carroll *et al*, 2006), this protein has been recently reassigned to the respiratory Complex IV (Balsa *et al*, 2012). Here, Markov clustering correctly placed NDUFA4 with Complex IV members COX5B and COX6C (Fig 1C and D). It should be noted, as multiple member complexes can be separated into smaller clusters during Markov clustering (Guruharsha *et al*, 2011), the spatial dimension of BN-PCP-SILAC allows the reconstruction of these larger complexes (Fig 1D). Furthermore, the observation of NDUFA4 migration as three distinct populations, which co-eluted with other known Complex IV members (Fig 1E and Table EV10), further supports the assignment of interactors and is consistent with Complex IV forming multiple supra-molecular associations (Vonck & Schafer, 2009).

**Apoptosis-induced changes in the mitochondrial and cytoplasmic interactomes in Jurkat cells**

Having established that PCP-SILAC can be applied to membrane and cytosolic interactomes alike, we then used these approaches to test

a key hypothesis in apoptosis: That apoptosis leads to broad alterations in protein complexes (Mahrus *et al*, 2008; Stoehr *et al*, 2013). Induction of the intrinsic apoptotic pathway has been extensively studied in the human Jurkat T cells (Scaffidi *et al*, 1998; Algeciras-Schimnich *et al*, 2002; Peter *et al*, 2007), where the aggregation of Fas leads to the permeabilization of the mitochondria and the activation of executioner caspases that orchestrate cellular destruction (Impens *et al*, 2010; Crawford & Wells, 2011). Although refinements to analytical approaches have greatly enhanced our knowledge of end-point caspase targets (Dix *et al*, 2008, 2012; Mahrus *et al*, 2008; Agard *et al*, 2012; Shimbo *et al*, 2012), their effect on the interaction landscape during apoptosis initiation is unknown.

Consistent with other studies, 4 h after Fas stimulation corresponded to the midpoint in the intrinsic apoptotic time course of Jurkat T cells (Na *et al*, 1996; Dix *et al*, 2008), with caspase target cleavage (e.g., PARP1; Nicholson *et al*, 1995) and DNA nicking (Appendix Fig S4A and B). We used BN- and SEC-PCP-SILAC to measure membrane and cytosolic interactomes, respectively, identifying 3,216 and 4,028 protein groups (Table EV11A and B). Among these groups, 2,779 yielded high-quality protein profiles, most of which were unique to either the membrane or the cytosolic interactomes (Fig 2A and Table EV12). From these proteins, a total of 7,502 and 5,102 individual fitted curves were mapped across three biological replicates in the cytoplasmic and organelle preparations, respectively (Table EV13A and B). Importantly, the utilization of our bioinformatics pipeline (Scott *et al*, 2015) enabled the re-alignment and quantitation of features across biological replicates overcoming variability resulting from independent fractionation experiments. Together, these findings were used to elucidate 30,789 (17,991 non-redundant) PPIs at a global precision of 68% (Tables EV14 and EV15). The segregation of mitochondrial and cytosolic complexes was nearly complete, with < 1% of the total observed interactions being shared (107 unique interactions, Fig 2B). We used Markov clustering, to assemble individual binary interactions into 2,591 and 1,037 discrete complexes across the cytosolic and membrane interactomes, respectively, with complex membership ranging from 2 to 61 members/components (Fig 2C and D, and Table EV16A–G). The connectivity of the observed networks for both the cytosol and membrane interactomes exhibited a strong linear trend (Appendix Fig S5A and B) typical of scale-free networks, albeit with the cytosolic interactome showing higher connectivity than the membrane interactome (7.02 members/complex vs. 3.65, Appendix Fig S6 and Table EV16A), consistent with other membrane interaction studies (Babu *et al*, 2012). Numerous proteins in the interactome were significantly changed in abundance ($P < 0.05$) by apoptosis (363 of 7,502 features in the cytosol, 147 of 5,102 features in the mitochondria; Fig 2E and F and Table EV13A and B). However, the vast majority of interactors,

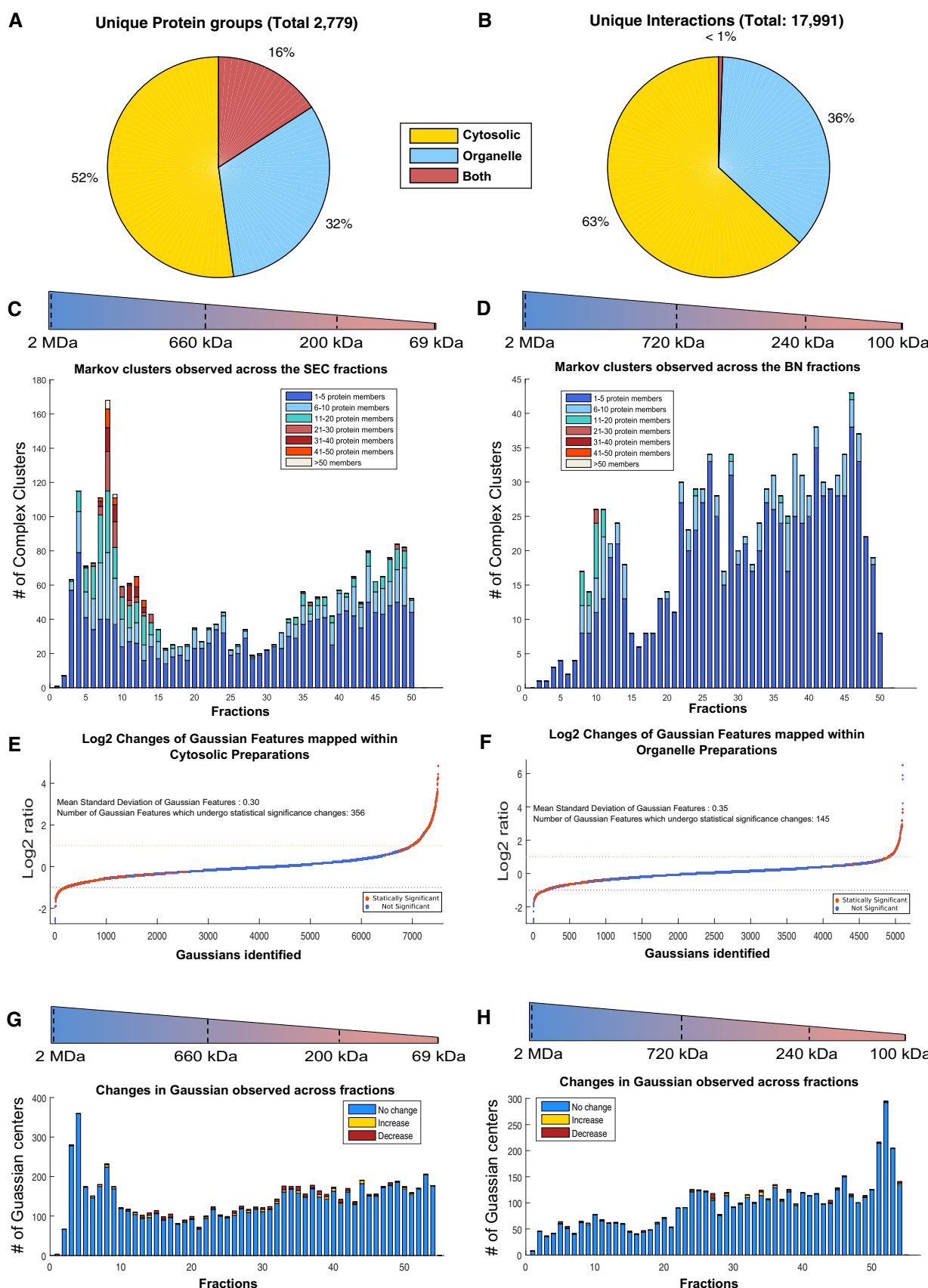

**Figure 2.**

**◄**

**Figure 2.  Alterations within the organelle and cytosolic interactomes in response to apoptosis.**

A        Overlap of protein groups identified in the PCP-SEC and PCP-BN interactomes, 16% overlap was observed at the protein level.

B        Overlap of protein interactions identified in the PCP-SEC and PCP-BN interactomes reveal that only a < 1% overlap was supporting the complementary nature of using two cellular preparations to explore the interactome.

C, D    Complexes observed within the cytosolic and membrane interactome based on Markov clustering mapped to position within the separation gradients. Size range corresponding to the calibrated molecular mass is given.

E        Observed changes in the interactome of cytosolic preparation across biological replicates, 7,502 Gaussian features were mapped across 55 fractions with a mean standard deviation of 0.297 resulting in the identification of 356 significant changes.

F        Observed changes in the interactome of mitochondrial/organelle preparation across biological replicates, 5,102 Gaussian features were mapped across 55 fractions with a mean standard deviation of 0.35 resulting in the identification of 145 significant changes.

G, H    Observed changes across the separation gradients of cytosolic and mitochondrial/organelle preparations. Changes are observed across nearly all fractions, yet the majority of mapped Gaussian features are unaffected by the initiation of apoptosis.

over 95%, were unaffected by the initiation of apoptosis confirming both the temporal dependence and specificity of the apoptotic program and robustness of cellular systems. This selectivity is important as it suggests that 4-h post-Fas stimulation is a suitable time point for exploring the early stages of apoptosis without the confounding influences of secondary events being invoked as part of the general cell disassembly response after a point of no return. In support of this, the interactions changed at 1 h Fas stimulation were virtually the same as untreated samples at 4 h (Appendix Fig S7).

Among the membrane and cytosolic interactomes significantly affected by apoptosis (Fig 2E–F), we observed redistribution of numerous proteins previously associated with Fas-mediated signaling. For instance, in the organelle preparations, we detected an apoptosis-induced increase in abundance of several complexes containing the initiator of mitochondrial permeabilization, BH3-interacting domain death agonist (tBid, P55957, Appendix Fig S8). Interestingly, only C-terminal tBID peptides (amino acids 72-195) were observed in cytoplasmic SEC-PCP-SILAC fractions, consistent with caspase-mediated processing of tBID at the previously reported site Asp[59] (Appendix Fig S8; Li et al, 1998).

Although the observed mitochondrial changes support a role for membrane interactome remodeling in apoptosis initiation, the degree of alteration in this interactome was far less pronounced than within the cytosol interactome (two-sample F-test for equal variances, P-value = $4.1689 \times 10^{-34}$). There was no obvious bias in the sizes of complexes that were affected by apoptosis (Fig 2G and H), but the proteins that were affected were highly enriched for caspase targets previously reported in the curated proteolysis database, Degrabase (Crawford et al, 2013) (P-value 0.0099506 for mitochondrial and $7.6779 \times 10^{-6}$ for cytosolic preparations, Table EV17). This could occur in one of three ways: proteolysis drove the rearrangement of the complexes, the rearrangement has led to their subsequent proteolysis or proteolysis, and complex rearrangement are two independent processes induced by apoptosis. Thus, the complementary analyses of cytosol and mitochondria demonstrate that although the interactome at 4-h post-Fas stimulation is predominantly unaffected, there are a few select complexes that change in abundance and/or composition during the early stages of apoptosis when the cell is already affected by the death stimulus and proceeding to full destruction.

Among the nearly 18,000 detected interactions, only 271 (1.5%) were "all-or-nothing," that is, observed in ≥ 2 replicates in only the stimulated or unstimulated state. The 34 such interactions in mitochondria (Table EV18A) showed no enrichment for any functional GO term, while the 237 gain/loss interactions detected in the

cytosolic interactome show enrichments in both interactions gained and lost (Table EV18B). Interactions gained in the cytosol were highly enriched for the "protein–DNA complex subunit organization" functional term (P-value $3.304 \times 10^{-5}$, Table EV18C), consistent with previous observation of an increase in DNA-related proteins (Wu et al, 2002) within the cytosol due to the disintegration of the nucleus, a hallmark of apoptosis (Appendix Fig S9). Interestingly, the interactions lost during apoptosis were enriched in multiple terms related to protein complex maintenance, such as "chromatin assembly or disassembly" and "cellular macromolecular complex assembly" (P-value 0.000128 and 0.006661, respectively, Table EV18C). This is consistent with the guiding hypothesis here that the initiation of apoptosis drives protein complex disassembly.

Due to the enrichment of terms related to protein complex maintenance and the previous association of DNA repair and transcription with caspase targets (Mahrus et al, 2008; Stoehr et al, 2013), we closely examined integrity of condensin and its interactions in apoptosis. Within our interactome, three known condensin I complex members were identified NCAPD2, NCAPH, and NCAPG (Q15021, Q15003, and Q9BPX3, respectively) as interacting partners (Fig 3A). The effect of Fas stimulation on all three proteins was nearly identical, each having a log$_2$ SILAC ratio of -1.3 (P-value $1.025 \times 10^{-6}$, $3.46 \times 10^{-4}$, $1.193 \times 10^{-5}$, Table EV13A). Within each biological replicates, all three proteins showed an identical response to treatment (Fig 3B), further supporting the determined association. The chromatograms of all three proteins were mapped across nearly all SEC fractions, leading to the conclusion that these complexes completely disintegrated in apoptosis, rather than simply transforming into a lower molecular weight complex (Fig 3C). This was also largely consistent within the peptide data underlying each protein chromatogram, although there were some obvious outliers (Fig EV3). Specifically, multiple peptides in the N-terminal section of NCAPH increased in abundance in fractions 36–41, while peptides in the C-terminal region were unaffected, suggesting that domains of this protein participate in different complexes and may correspond to stable cleavage fragments previously observed in apoptosis (Dix et al, 2008; Stoehr et al, 2013).

Proteolysis is an important post-translational modification that can terminate protein function through their degradation, whereas proteolytic processing specifically alters protein structure, localization, and function through precise and measured cleavage events leading to stable cleavage fragments (Lange & Overall, 2013; Marino et al, 2015). To further explore whether protein fragments generated after cleavage accumulate or are depleted, and thus behave differently than suggested by the whole protein averages, we employed

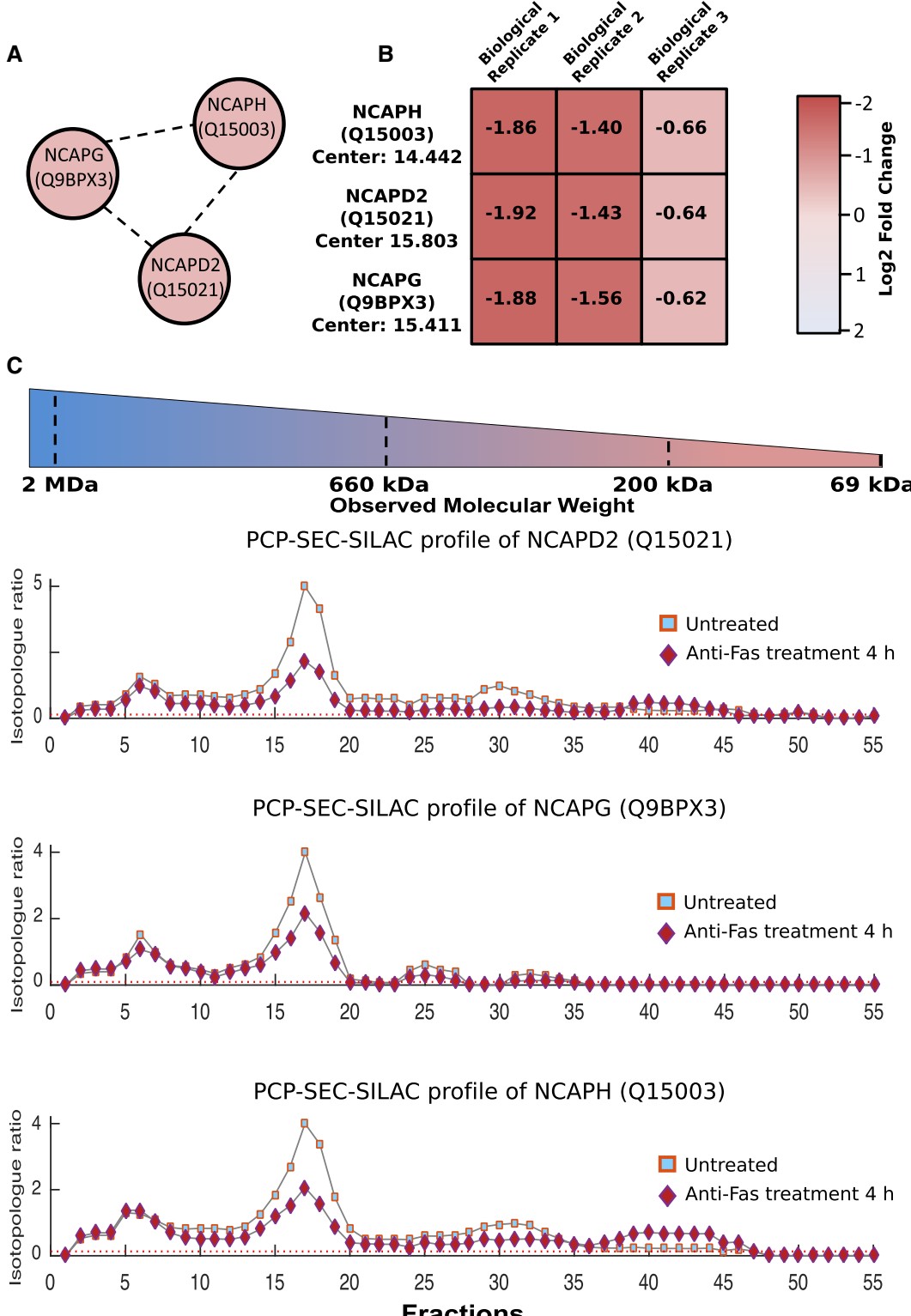

**Figure 3.  Changes in the condensin I complex in response to apoptosis.**

A   All three observed members of condensin I, NCAPD2, NCAPH, and NCAPG were correctly assigned as interacting partners using PCP-SEC-SILAC.

B   Across the three biological replicates, all three members of condensin I undergo identical decreases in response to treatment with a mean $\log_2$ fold change of −1.3 observed in response to apoptosis.

C   Protein profiles of condensin I members NCAPD2, NCAPH, and NCAPG within biological replicate 3 demonstrating the near identical alteration in the observed protein profiles in response to the initiation of Fas-mediated apoptosis.

the Shannon index (Shannon & Weaver, 1949; Bent & Forney, 2008) to globally assess peptide chromatogram diversity and evenness. The Shannon index allows global identification of proteins whose measured peptides have a higher/lower "unevenness" based on SILAC ratios observed across the protein coverage of a given PCP fraction. In other words, this statistical approach seeks to identify proteins with high protein coverage that, in response to treatment, display a high degree of variability across the SILAC values of individual peptides. Such a case would be consistent with formation of stable protein fragments, as has been observed previously following proteolytic processing by a peptide-mapping approach (Dean & Overall, 2007). Based on their Shannon indices, 1.9% of proteins in the mitochondrial and 2.6% of proteins in the cytosolic interactomes were uneven (Fig 4A and Table EV19A and B). This small percentage of proteins further indicates high selectivity of the early apoptotic response and was similar to that affected by Fas stimulation at the interactome level. Consistent with these phenomena being linked, proteins whose interactions were altered by apoptosis also tended to display unevenness across their peptides (*P*-value $3.86 \times 10^{-22}$ within the cytosol and $3.28 \times 10^{-12}$ within the mitochondria, Table EV19C and D). Given this overlap, it is unsurprising that similar functional enrichments were observed (Fig 4B); for example, proteins possessing high unevenness in the cytosolic interactome were functionally enriched in protein–DNA complex subunit organization and cytoskeleton terms (*P*-value $1.25 \times 10^{-15}$ and $2.51 \times 10^{-5}$, Table EV19C).

One interesting example of such a protein is the cytoskeletal protein filamin-B (O75369, Fig 4C). The individual peptides identified from filamin-B revealed that only part of the protein responded to apoptosis initiation, but clearly this could not happen if the protein was intact. The peptide chromatograms revealed two regions of the protein, segregated somewhere between amino acids 1,600 and 1,900, that increased in abundance during apoptosis initiation. As the C-terminus of filamin-B mediates a range of protein interactions, especially with other components of the cytoskeleton (Stossel *et al*, 2001; Takafuta *et al*, 2003), our data suggest that proteolytic alterations may lead to changes in protein interactions of filamin-B. This example and the other 2% of proteins with high unevenness suggest that proteolytic events either cause or are the effect of the changing interactome. Therefore, we next addressed this key question.

## Interactome rearrangement during apoptosis does not correlate with caspase-mediated proteolysis

The degradative role of caspases during the later stages of apoptosis has been extensively characterized, with over 8,000 cleavage sites identified in 1,700 proteins in response to a range of stimuli (Dix *et al*, 2008; Crawford & Wells, 2011; Crawford *et al*, 2013; Wiita *et al*, 2013)—these events were hypothesized to be the driving force behind disassembly of the interactome (Mahrus *et al*, 2008; Stoehr *et al*, 2013). However, the role of caspases in altering protein–protein interactions by proteolytic processing of complex components during early stages of apoptosis has never been demonstrated. Therefore, to assess the global effect of proteolytic processing, particularly by caspases, on the apoptotic interactome, we employed TAILS (Kleifeld *et al*, 2010), for the identification and quantification of protein N-termini, both

natural as translated, and the neo-N-termini resulting from proteolysis.

By performing TAILS only on the ≥ 100-kDa filtrate isolated under non-denaturing conditions, we identified specific cleavage sites in the isolated protein complexes and were also able to link any observed cleavage events in a protein to corresponding changes in its interactions. N-termini were quantified by SILAC in the untreated *vs.* Fas-stimulated cells in the presence or absence of the caspase inhibitor Z-vad-FMK, pre-incubated for 1 h before apoptosis initiation. TAILS identified 1,892 unique N-termini corresponding to 857 proteins across three biological replicates in these protein complexes (Table EV20). 86% of identified peptides corresponded to mature true N-terminal peptides (naturally acetylated or dimethylated in TAILS) (Fig EV4A). Most of the N-termini were in proteins found within the cytosolic interactome (674 out of 857), whereas a smaller fraction of proteins (183 of 857) were unique to the N-terminome (Fig EV4B and C). Of the 674 shared proteins, 608 proteins were quantified in both, thus providing a simultaneous analysis of the N-terminome and the interactome during apoptosis initiation.

Consistent with previous observations (Mahrus *et al*, 2008; Stoehr *et al*, 2013), apoptosis triggered widespread alterations in the N-terminome, revealing a bimodal distribution of N-termini ratios in response to Fas-induced apoptosis (Fig 5A, brown bars). Such a distribution reflects two distinct proteolytic processes: protein degradation (SILAC ratios centered at $\log_2 -0.9$ and ranging between $-4$ and 1, $n = 753$) and proteolytic processing generating stable cleavage products (SILAC ratios centered at $\log_2 1.6$ and ranging between 1 and 5, $n = 411$) (Fig 5A). Consistent with a prominent role of caspases in apoptosis, Fas treatment induced multiple caspase cleavage events exhibiting the typical caspase motifs DEVD↓G (Wejda *et al*, 2012) and DEXD↓G (Thornberry *et al*, 1997) ($n = 77$ increasing and $n = 53$ decreasing in response to treatment, *P*-values $4.17 \times 10^{-10}$ and 0.009; Fig 5B and C, and Table EV21). For instance, we detected a fourfold decrease (*P*-value = 0.15442) in the levels of the prime side cleavage peptide D.$^{1046}$SITNQIALLEAR$^{1057}$ of the known caspase substrate SMC4 (E9PD53) (Stoehr *et al*, 2013) and a 2.2-fold decrease (*P*-value 0.05024) in D.$^{72}$AALAVLEDR$^{180}$ terminus of proteasome subunit beta type-10 (P40306) (Gray *et al*, 2010), indicative of degradation of this fragment after caspase cleavage.

In cells pre-treated with the Z-vad-FMK caspase inhibitor prior to induction of apoptosis, we detected 157 termini that were stabilized relative to apoptosing cells (Fig 5A, blue vs. orange bars, Fig EV4D and E, and Table EV22). Stabilized N-termini could be grouped into three subpopulations: cleavage events C-terminal to aspartic acid, C-terminal to arginine, and natural N-termini (Appendix Fig S10A). Motif analysis of neo-N-termini peptides cleaved C-terminal to aspartic acid suggests that most of these resulted from caspase activity (Appendix Fig S10B). However, not all Asp-directed cleavage events were Z-vad-FMK-sensitive, particularly for neo-termini resulting from proteolytic processing (as opposed to degradation; Fig 5B vs. 5C). This suggests that other proteases are involved, either Z-vad-FMK-insensitive caspases (Garcia-Calvo *et al*, 1998) or in alternative protease families.

The SILAC ratios for cleavage events C-terminal to aspartic acid form a bimodal distribution, with the two populations corresponding to peptides that decrease (distribution centered at $\log_2 -0.93$, $n = 53$ below $\log_2 -1$) and peptides that increase (distribution centered at $\log_2 0.57$, $n = 77$ above $\log_2 1$, Appendix Fig S10C) in response to

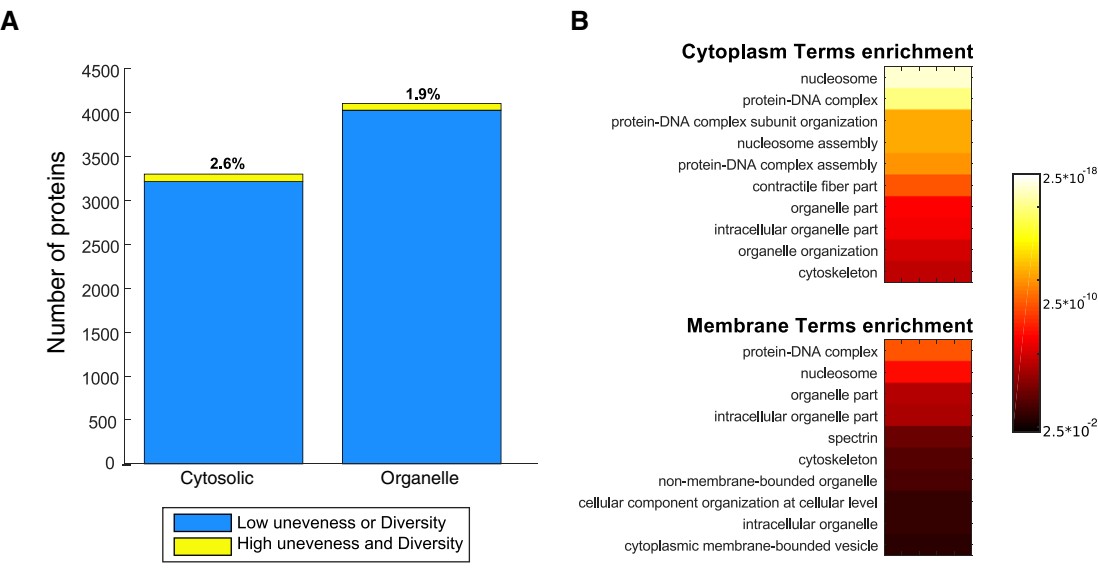

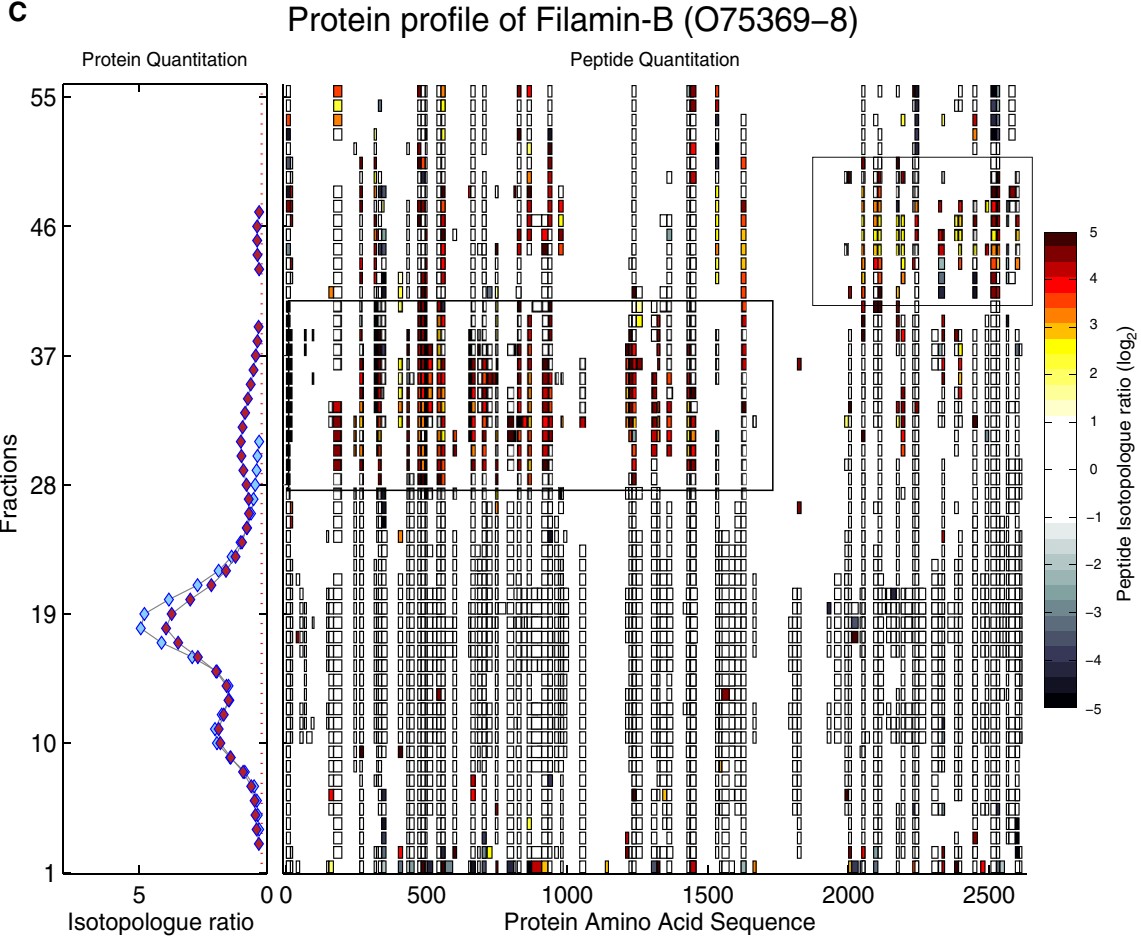

**Figure 4.  Identification of alteration in the peptide measurements within PCP fractions.**

A  Using the Shannon index, the evenness of peptide measurements was assessed in the cytoplasmic and organelle interactomes.

B  Enrichment analysis of protein groups observed to contain uneven peptide measurements shows that multiple terms are highly enriched in both cytoplasmic and membrane interactome.

C  Example of detected unevenness within peptide measurements in response to apoptosis for the protein filamin-B. Within the lower PCP-SILAC fractions, Gaussian features were detected which at the peptide level correspond to specific subregions of the protein that increases in response to apoptosis.

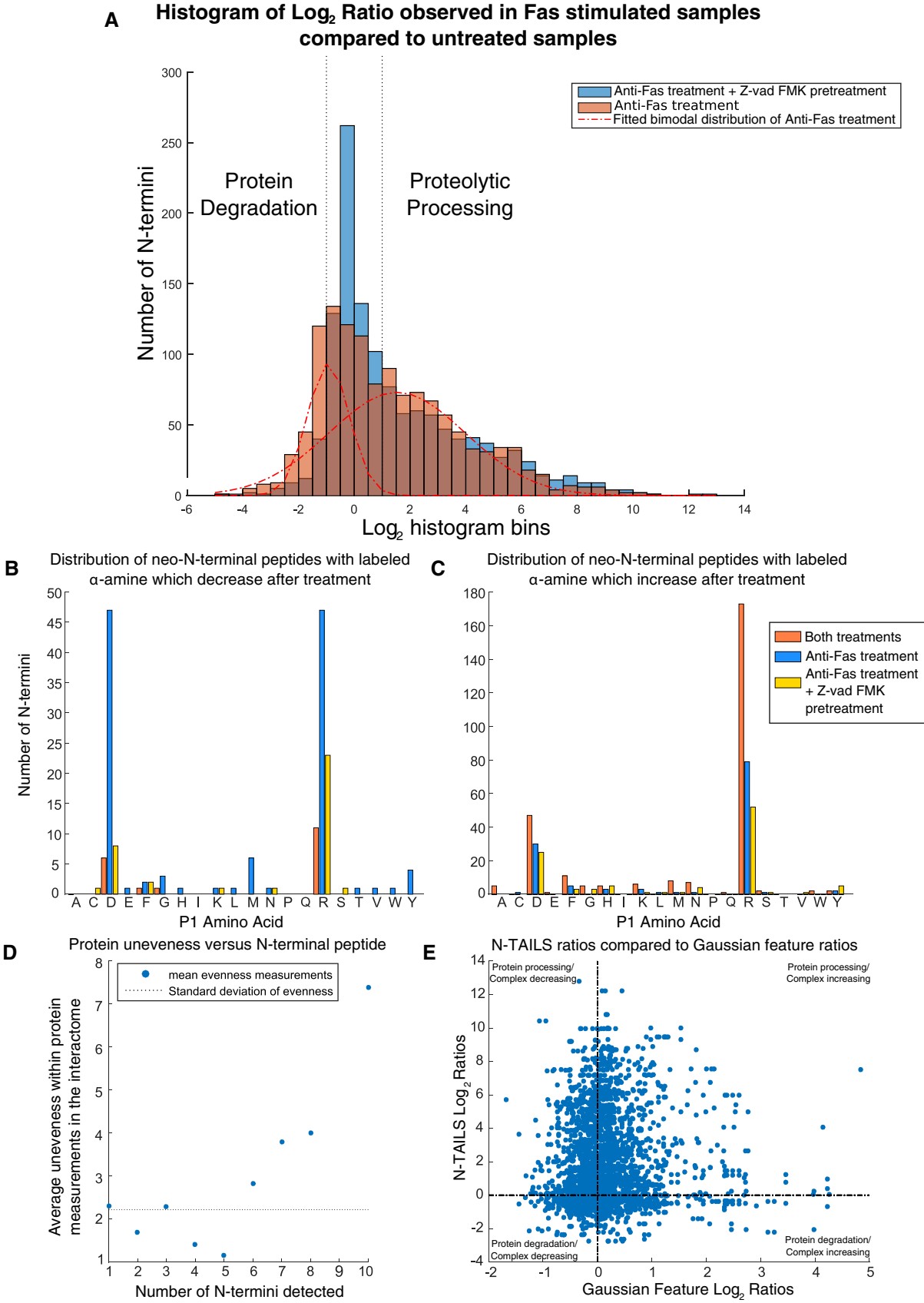

**Figure 5.**

◀

**Figure 5.    N-terminal peptide enrichment analysis of Fas-induced apoptosis.**

A    Histogram distribution of the ratio of observed N-terminal peptides after Fas treatment and Fas treatment with pre-treatment with caspase inhibitor Z-vad-FMK. The bimodal distribution of N-termini ratios ($r^2$ = 0.975) observed for Fas treatment is shown with a similar distribution fitted for Fas treatment with the caspase inhibitor Z-vad-FMK (Appendix Fig S14F). The position of $\log_2 1$ and $\log_2 -1$ is denoted with dotted black lines.

B, C    The distribution of the P1 amino acid identified for N-termini that was determined to be (B) below $\log_2 -1$ and (C) above $\log_2 1$ are shown. The number of neo-N-terminal peptides quantified below $\log_2 -1$ and above 1 are shown grouped according to the mapped P1 amino acids. Fas treatment without caspase inhibition is denoted by the blue bar and Fas treatment with the caspase inhibitor Z-vad-FMK denoted by yellow with the N-terminal peptides showing identical changes within each condition shown in orange.

D    The observed unevenness of proteins in response to apoptosis compared to the number of N-termini detected reveals as the number of N-termini peptides detected increases the average observed unevenness at the interactome level also increases.

E    Comparison of the TAILS ratio versus Gaussian ratios upon FAS treatment vs untreated samples for all combinations of observed TAILS and Gaussian ratios showing only a poor correlation (Spearman correlation: 0.21) between the two values supporting these events being unlinked.

Fas stimulation. These two populations of neo-N-termini revealed striking differences in cleavage consensus: Whereas the suppressed population was enriched in the typical executioner caspases motif, the increasing population had none of the acidic residues typically found in the P4-P2 positions of caspase motifs (Fig EV5). As further evidence that these were not caused by typical executioner caspases, the increasing cleavage events C-terminal to aspartic acid were insensitive to inhibition with Z-vad-FMK (Tables EV20 and EV22). The TopFIND database, which also integrates MEROPS data (Rawlings *et al*, 2012; Fuchs *et al*, 2013; Fortelny *et al*, 2015), suggested granzyme B or non-conventional caspases as the most likely protease responsible for these cuts (Van Damme *et al*, 2009, 2010b). Previous reports have suggested the absence of granzyme B within Jurkat cells (Froelich *et al*, 1996; Sedelies *et al*, 2004) and consistent with these western analysis shows the lack of detectable granzyme B in treated or untreated Jurkat cells (Appendix Fig 11A). Enzymatic cleavage assays also show the lack of proteolysis inhibition in Fas-treated Jurkat cells in response to pre-treatment with the granzyme B inhibitor Compound 20 (Willoughby *et al*, 2002) and only partial reduction in inhibition with Z-vad-FMK or a combination of Z-vad-FMK and Compound 20 (Appendix Fig 11B). These findings support the observed cleavage events be derived from or non-conventional caspases or alterative proteases. Consistent with the initiation of alterative proteases in response to apoptosis, many cleavage events are observed after an arginine (*n* = 462, Fig 5B and C). Analysis of these cleavages revealed the enrichment of negative residues at the P3 and/or P4 positions supporting the presence of substrate specificity within these cleavage event (Fig EV5). Combined these findings suggest that, whereas executioner caspases are important, alterative enzymes also play a role in the proteolytic processing at early stages of apoptosis in Jurkat cells.

The significant overlap of > 600 proteins between the N-terminome and the interactome allowed us to examine the mechanistic basis for the high unevenness observed for some proteins in our interactome (Fig 4). Interestingly, as the average number of neo-N-termini detected from a protein increases, so does the average unevenness (Fig 5D). Although this supports that unevenness in the interactome correlates with the number of neo-N-termini, the changes in the N-termini were poorly correlated with the changes in the interactome (Spearman correlation: 0.21, Fig 5E). This lack of correlation supports the conclusion that the changes in the neo-N-termini are not tightly linked to changes observed within the interactome. That is, cuts by caspases do not break up protein complexes; rather they *follow* remodeling of protein complexes, possibly by gaining access to cleavage sites otherwise masked or stabilized by protein–protein interactions. Nonetheless, we detected a limited number of cases where

proteins were cleaved and their interactions altered (*n* = 147 out of 2,794 possible combinations, Fig 5E). For example, the N-termini data for filamin-B (O75369) revealed multiple internal neo-termini, which increased in apoptosis in a caspase-independent manner (Table EV20), supporting the formation of cleaved proteoforms suggested by the PCP profiles (Appendix Fig S12). These N-terminal data are consistent with the filamin-B PCP profiles, suggesting that the cleaved proteoforms participate in different interactions. Thus, these data provide experimental evidence that some proteolytic products observed in apoptosis are not only stable (Dix *et al*, 2008, 2012), but also form new complexes with other proteins.

While these data argue against the hypothesis that caspase cleavage causes complex disassembly, identification of cleavage events by TAILS is dependent on the resulting termini being of sufficient length and composition to be identified by mass spectrometry. To assess the extent of processing, we sought to explore whether specific proteins of interest undergo cleavage on the same time scales and magnitude as alterations in the interactome. To achieve this, three proteins whose interactions changed in response to apoptosis and that have previously been confirmed as caspase targets (CDC42, TACC3, and NCAPH; P60953, Q9Y6A5, and Q15003; Crawford *et al*, 2013; Stoehr *et al*, 2013; Tu & Cerione, 2001) were further examined. Protein profiles of all three proteins exhibited dramatic changes 4 h post-Fas (Appendix Fig S13 and Table EV13A), yet only modest evidence for degradation or changes in protein abundance could be observed at the protein level (Fig 6A). Interestingly, even in the condensin I subunit NCAPH, that is known to undergo caspase-mediated processing (Lai *et al*, 2011), the observed processing was modest at few percent of the total protein pool (Fig 6A). These findings demonstrate that the magnitude of the interactome changes is far greater than at the level of changes in protein abundance or processing and suggests that the loss of these three proteins from the interactome, at least at the population level, is not due to removal by degradation.

To further complement these findings, confocal microscopy was utilized to assess the effect of apoptosis initiation on both the protein level and co-localization status at an individual cell level of CDC42 and its interaction partner IQGAP1. Consistent with previous reported, CDC42 interacted with IQGAP1 (Bashour *et al*, 1997) under untreated conditions (Fig 6B, Spearman rank correlation coefficients 0.8619, Table EV23), yet under Fas treatment, we noted cell-specific loss of CDC42 only within cells showing loss of nuclei morphology and mitochondrial membrane potential (Fig 6C, *P* = 0.0003, Mann–Whitney *U*-test, Table EV23). This loss of signal for CDC-42 supports degradation of CDC42 in response to apoptosis (Tu & Cerione, 2001) yet only within cell showing morphological

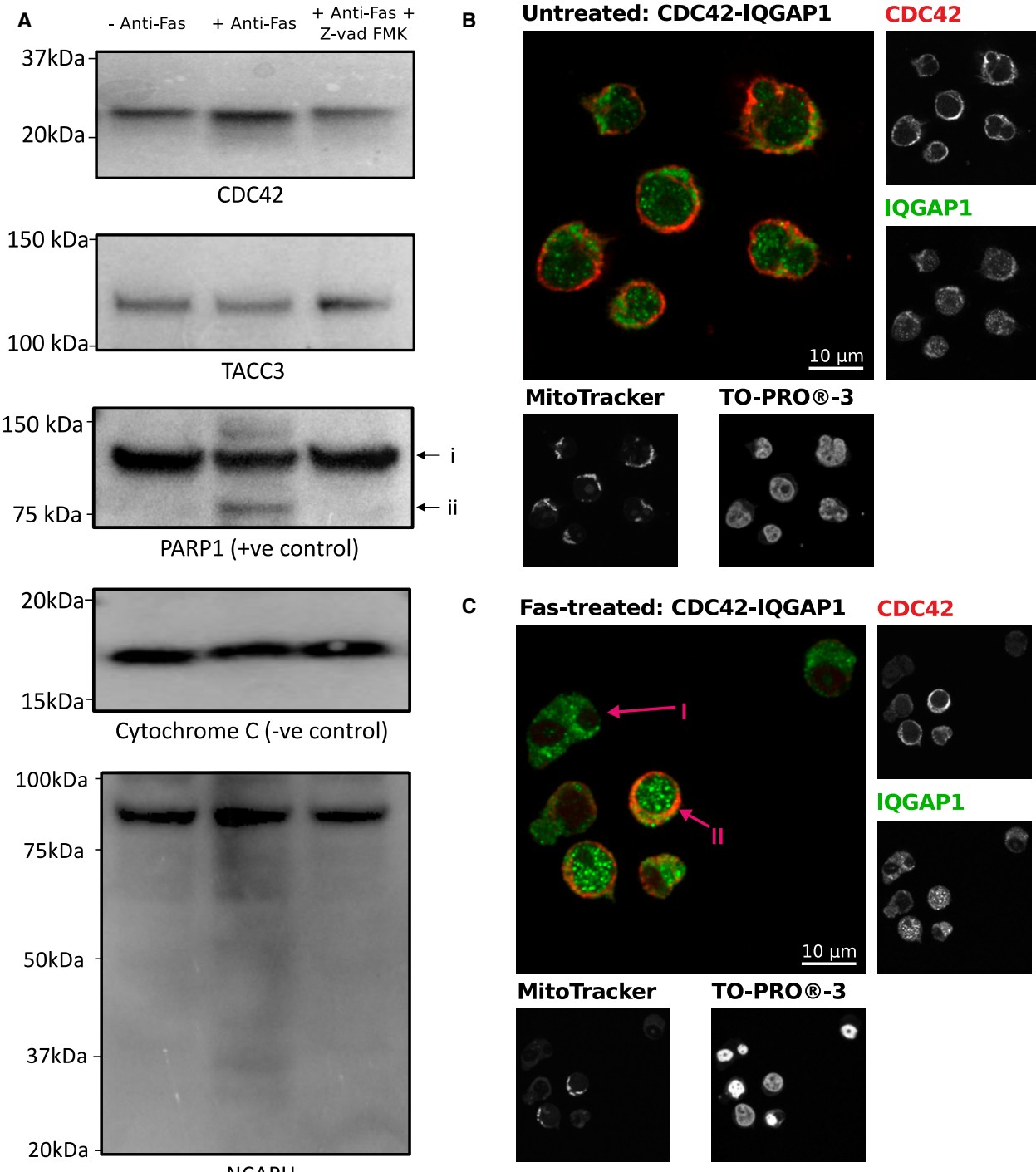

**Figure 6.  Western blot-based and co-localization investigation of protein-specific proteolysis of known caspase targets which undergo changes in the interactome.**

A    Western blotting analysis of TACC3, CDC42 and NCAPH with CYC1 (cytochrome c1) and PARP1 as controls. Lanes correspond to: (1) with no treatment, (2) in response to 4-h Fas treatment, and (3) in response to 4-h Fas stimulation with prior treatment with Z-vad-FMK. In response to treatment, TACC3, CDC42 and NCAPH do not undergo detectible cleavage. Examination of the low mass region of NCAPH supports some, albeit very minor, detectible proteolysis in a caspase-dependent manner after 4-h Fas stimulation.

B, C   Co-localization analysis of CDC42 and its interaction partner IQGAP1 without (B) and with 4-h Fas stimulation (C). Anti-CDC42 and anti-IQGAP1 staining shows co-localization of CDC42 and IQGAP1 within cells which have maintained mitochondrial membrane potential, as determined by the retention of MitoTracker, and non-condensed nuclei as determined by TO-PRO-3 iodide. Arrow (I) highlights an example of apoptosis-committed cell showing loss of CDC42, loss of mitochondrial membrane potential, and condense nuclei. Arrow (II) highlights an example of a pre-apoptotic cell which has maintained CDC42, mitochondrial membrane potential, and nuclei is non-condensed.

changes consistent with total commitment to apoptosis and suggests within the majority of cells CDC42 co-localization does not change in response to early Fas treatment. Thus, irrespective of caspase target enrichment, other factors are drivers of the dramatic changes seen in the interactome independent of proteolytic events at least at the early stages of apoptosis initiation.

## Discussion

Here we have mapped the cytosolic and organelle membrane interactomes, as well as proteolytic cleavage events, of Jurkat cells undergoing early stages of apoptosis and used this information to address the key assumption in the apoptosis field: that the caspase cascade triggers dissociation of the cellular machinery. We report 17,991 non-redundant interactions (6,651 membrane and 11,447 cytosolic interactions), 271 of which were altered at 4 h after apoptosis induction. Although the affected part of the interactome was highly enriched for previously described caspases targets, these cleavages were not detected at the early time point tested here. Furthermore, the observed caspase cleavages did not necessarily correspond to altered interactions (Fig 5). This observation, in conjunction with the low correlation between changes within the interactome and the N-terminome, suggests that, at least at a system level, most of the dramatic interactome changes are independent of and preceding caspase-mediated proteolytic processing.

Our data further demonstrate that apoptosis is not a wholesale disassembly of the cellular machinery: Rather, a number of well-defined cytosolic complexes are among the first targets, whereas membrane complexes are largely spared. As the disassembly of these complexes occur early during apoptosis, it is tempting to speculate that the loss of these complexes drivers apoptosis akin to the cleavage of caspase targets such as ICAD (Sakahira *et al*, 1998) and BID (Li *et al*, 1998). However, the essentiality of the loss of these complexes to the initiation of apoptosis has yet to be determined. The most fundamental observation of this work is that proteolysis is not the trigger for protein complex disassembly and does not account for the magnitude of the alterations observed in the interactome. Previous observations had noted that, even in the constant presence of pan-caspase inhibitors, morphological and phenotypic changes occur upon the initiation of apoptosis (Xiang *et al*, 1996; McCarthy *et al*, 1997; Bortner & Cidlowski, 1999; Johnson *et al*, 2000), suggestive of other mediators in addition to caspases. Our study addresses this by measuring the generation of neo-N-termini, in parallel to monitoring interactome rearrangements. Of 608 proteins for which at least one N-terminus and one cytosolic interaction were observed, most termini changed little in response to apoptosis, even in cases where the complexes had already completely disassembled (Fig 5E). At the protein level, we see that even in known caspase targets (CDC42, NCAPH, and TACC3), the amount of degradation can be minimal yet the changes to the interactome dramatic. This finding is consistent with earlier work by Stoehr *et al* (2013) who noted that in response to the induction of apoptosis, the effect on the total protein level can be small, yet fragments are readily observable. Although our and Stoehr *et al* observations support minimal proteolysis during the early stages of apoptosis, these data do not provide complete details on its functional consequences: It is

currently unknown whether complex disassembly can be triggered by cleavage of any individual complex member or whether it requires a critical mass of cuts in several complex members. These dynamics would be highly unique for each specific complex and defined by its features, such as number of subunits, affinities and kinetics of their interaction, specific location of a cleavage(s), and its penetration within the total pool of that specific protein subunit. Furthermore, as stable cleavage products can have dominant negative effects on the cell or form new unique protein complexes, their formation, even at low levels, can lock the cell into a path of no return, descending to cell death. Conversely, single cleavage events could manifest in serious functional consequences without apparent changes in the interactome: The complex may still be intact but non-functional when missing a crucial domain or even few residues.

Beyond just the clear caspase targets, that is, those sensitive to Z-vad-FMK, we also identified many alterative proteolytic events, particularly C-terminal to Arg residues and Z-vad-FMK-insensitive cleavage at Asp. Interestingly, these cleavage events conform to motifs consistent with those of granzyme activities (Figs 5B and C, and EV5; Van Damme *et al*, 2010a,b), yet Jurkat cells have been suggested to lack granzyme granules (Sedelies *et al*, 2004). While Jurkat cells can be stimulated to induce components of the granzyme granules, such as granzyme B (Huang *et al*, 2006; Smeets *et al*, 2012), under our experimental conditions, we saw no evidence for granzyme B expression or activity (Appendix Fig S11) suggesting the Z-vad-FMK-insensitive cleavage at Asp events are not granzyme B-derived and are products of yet another protease(s). This suggests the Z-vad-FMK-insensitive Asp cleavage events originate from either another protease class or non-executioner caspases. As Z-vad-FMK inhibition is not uniform (Garcia-Calvo *et al*, 1998) and multiple caspases are activated during Fas-mediated apoptosis, these Z-vad-FMK-insensitive cleavage are potentially derived from non-conventional caspase such as caspase-2 (Lavrik *et al*, 2006). In addition to the caspase cleavage events, others have also noted non-caspase proteolytic events with similar kinetics as caspases during apoptosis, including those that occur C-terminal to arginine (Wiita *et al*, 2013). Thus, our data support that alterative proteases to the executioner caspases play a key role in initiating apoptosis (Weis *et al*, 1995). Irrespective of the origins of the observed cleavage events, the availability of both interactome and N-termini data allows us to test, in an unbiased way, the general assumption that proteolytic cleavage events during apoptosis trigger the disassembly of the cellular machinery. This hypothesis, although intuitive, is not supported by our findings and is a key conclusion of the present study. This suggests that additional steps in the initiation of apoptosis must occur independently of caspase-mediated and non-caspase-mediated proteolysis in order to achieve the rapid commitment to cellular death.

The mitochondrial membrane interactome was, surprisingly, largely unaffected 4 h after Fas stimulation. Despite this, changes known to occur prior to the commitment toward apoptosis, such as the insertion of tBID into the membrane, were readily observable (Appendix Fig S8). However, known targets of caspase-3, such as NADH dehydrogenase (NDUS1, P28331; Ricci *et al*, 2004) of Complex I, did not undergo dramatic changes under our experimental conditions (Appendix Fig S14) supporting that these protein complexes are unaffected by the priming of the cell toward

apoptosis. This is consistent with previous reports that NDUS1 cleavage occurs after the release of cytochrome C when cells are fully committed to apoptosis (Ricci *et al*, 2003, 2004).

The commitment toward apoptosis is a highly regulated event (Elmore, 2007; Li & Dewson, 2015), with increasing evidence suggesting that multiple signals are required to safeguard cells from unintentional death (Morgan *et al*, 2015). It now appears that disassembly of the protein machinery of the cell is an additional step and one that is independent, at least partially, of the caspase cascade. Protein complex dissociation prior to cleavage is conceptually appealing, as it would provide a further mechanism for fine-tuning apoptosis through reversible disassembly/assembly of protein complexes as opposed to irreversible proteolysis. In fact, this observation reconciles multiple findings within the field and supports a model of apoptosis initiation that allows rapid commitment to cell death via multiple, simultaneous caspase cleavage events: Recent single cell studies have shown that on commitment of cells to apoptosis, an all-or-none switching phenotype is observed where effector caspase substrates are rapidly processed (Albeck *et al*, 2008a,b). Consistent with this, we observe the loss of the known caspase target CDC42 only within cell showing known morphological changes consistent with commitment of cells to apoptosis (Fig 6B and C). The dissociation of complexes prior to degradation, as observed here, would expose all members of a complex to degradation, a phenomenon which has been noted for condensin (Stoehr *et al*, 2013). Although the cause of these dramatic interactome changes is unknown, the scope of complex remodeling suggests that the mediator acts in a rapid, pleiotropic manner consistent with initiation by a protein modification. Intriguingly, modifications such as phosphorylation and glycosylation have both been shown to augment apoptosis (Zhu *et al*, 2001; Dix *et al*, 2012), yet the potential connection of these protein modifications to interactome rearrangement remains to be tested. The modulation of protein interactions in apoptosis may also explain numerous nearly paradoxical findings within the field, such as the observed kinetics of caspase cleavages, which may differ by nearly 500-fold for different substrates in one cell type (Agard *et al*, 2012) and also vary with different apoptotic stimuli and cell types (Shimbo *et al*, 2012).

In summary, we have adapted the PCP-SILAC methodology to characterize membrane interactions using BN-PAGE allowing the first high-throughput assessment of the membrane interactome changes. Using PCP-SILAC, we investigated the initiation of Fas-mediated apoptosis and generated the largest dynamic interactome to date. Similarly to the proteome abundance studies where only modest changes in the proteome have been observed, both for Fas-mediated apoptosis (Thiede *et al*, 2001; Mahrus *et al*, 2008) and alterative apoptotic stimuli (Wiita *et al*, 2013), we detected few dramatic alterations in the interactome upon apoptosis initiation. These changes frequently involved known caspase targets within the mitochondria and cytoplasm, yet the cytoplasmic interactome was significantly more affected. Activity of at least two protease classes, caspases and currently anonymous proteases cleaving at Asp and Arg, lead to the generation of multiple neo-N-termini, yet these N-terminal changes did not correlate with interactome alterations. Together, our findings suggest that large alterations within the interactome seen in early stages of apoptosis occur largely independent of caspase activity.

# Materials and Methods

### Cell culture and induction of apoptosis

Jurkat cells were grown in RPMI-1640 supplemented with 10% fetal bovine serum (FBS) (Invitrogen, Burlington, ON, Canada), 100 units of penicillin/streptomycin (Gibco/Invitrogen), and split 1:5 24 h prior to the induction of apoptosis. Cells were harvested and placed in fresh pre-warmed media at a density of $1 \times 10^6$ cells/ml for 2 h prior to the initiation of treatment. Apoptosis was initiated using anti-Fas monoclonal antibody (CH11, EMD Millipore) at 250 ng/ml for 0, 0.5, 1, 2, 4, 8, 16, or 24 h. At the desired time point, cells were collected, washed twice with ice-cold PBS, and either snap frozen with liquid nitrogen and stored at −80°C or fixed in 1% (w/v) paraformaldehyde in phosphate-buffered saline (PBS) for 15 min, washed with PBS, and stored in ice-cold 70% (v/v) ethanol at −20°C. For the inhibition of caspase activity, the pan-caspase inhibitor Z-vad-FMK was added at 20 μM for 1 h prior to initiation of anti-Fas treatments.

SILAC labeling of Jurkat cells was accomplished as previous described (Stoehr *et al*, 2013). Briefly, RPMI-1640 (Lys/Arg$^{-/-}$) was supplemented with 10% dialyzed FBS (Invitrogen, Burlington, ON, Canada), 1× penicillin/streptomycin, and combinations of the following lysine and arginine isotopologues: for "light" or "L"-labeled cells L-arginine (34 mg/l) and L-lysine (73 mg/l) (Sigma-Aldrich, Oakville, ON); for "medium" or "M"-labeled cells $^{13}C_6$-L-arginine (35 mg/l) and $D_4$-L-lysine (74.8 mg/l); and for "heavy" or "H"-labeled cells $^{13}C_6{}^{15}N_4$-L-arginine (35.8 mg/l) and $^{13}C_6{}^{15}N_2$-L-lysine (76.6 mg/l) (Cambridge Isotope Laboratories, Andover, MA, USA). Cells were split 1:4 into the three SILAC media formulations and passaged five times for complete replacement of labeled amino acids. For each condition within biological replicate, $1 \times 10^8$ cells were used with cells harvested and placed in fresh pre-warmed media at a density of $5 \times 10^6$ cells/ml for 2 h prior to the initiation of treatment.

### Flow cytometry

Flow-based monitoring of DNA cleavage was accomplished using the APO-BrdU TUNEL Assay kit (Invitrogen). Fixed cells were washed and prepared for DNA nicking/propidium iodide labeling according to manufactures instructions. Cells analyzed using a FACSCalibur system (Becton Dickinson, San Jose, CA), and data analyzed using FlowJo (v8.7).

### Western blot

Cells were lysed on ice in freshly prepared RIPA lysis buffer [150 mM NaCl, 1.0% IGEPAL® CA-630, 0.5% sodium deoxycholate, 0.1% SDS, 50 mM Tris, pH 8.0, supplemented with Complete protease inhibitor cocktail without EDTA (Roche)]. Samples were clarified by centrifugation at 20,000 relative centrifugal force (r.c.f.) for 20 min at 4°C, and protein concentrations were determined using a BCA protein assay. Twenty micrograms of protein lysates was used for immunoblotting, with the proteins resolved on 4–12% NuPAGE Bis-Tris Gels (Invitrogen) run with 1× MOPS buffer (Invitrogen) for 45 min at 200 V. Proteins were transferred onto PVDF membranes using a wet-blotting system for 1 h at 400 mA. PVDF

was blocked overnight in 5% skim milk powder in PBS-T and then probed with primary and then secondary antibodies in 1% skim milk in PBS-T, with three washes of PBS-T between probes. Primary antibodies against PARP (# 9542, Cell Signaling), TACC3 Antibody (D-2, sc-48368, Santa Cruz), CDC-42 (P1, SC-87, Santa Cruz), cytochrome C (7H8.2C12, Abcam), NCAPH (HPA002647, Sigma-Aldrich), and granzyme B (# 4275, Cell Signaling) were utilized at 1:5,000. Blots were developed with HRP-conjugated secondary antibody diluted 1:10,000 in PBS-T using chemiluminescence method (ECL) using either Hyperfilm detection (GE Healthcare Life Sciences) or digital detection using a Gel Doc™ XR+ System (Bio-Rad).

### Preparation of cytoplasmic complexes for PCP-SILAC separation

Cell lysis and size exclusion chromatography were performed as described previously (Kristensen *et al*, 2012), with minor modifications. Briefly, after treatment cells were immediately harvested by centrifugation at 700 r.c.f. for 5 min, 4°C, and washed three times with ice-cold PBS. Harvested cells of the same SILAC label were pooled and re-suspended in 2 ml of ice-cold size exclusion chromatography (SEC) mobile phase [50 mM KCl, 50 mM NaCH$_3$COO, pH 7.2, containing Complete protease inhibitor cocktail without EDTA (Roche), and additional phosphatase inhibitors (5 mM Na$_4$P$_2$O$_7$, 0.5 mM sodium pervanadate)]. Cells were lysed by 200 strokes with a Dounce homogenizer, and insoluble material was removed by ultracentrifugation at 100,000 r.c.f. for 15 min at 4°C and lysates concentrated using 100,000 Da molecular weight cutoff spin columns (Sartorius Stedim, Goettingen, Germany). Equal amounts of protein from heavy-labeled and medium-labeled cells lysates were combined and immediately loaded onto a chromatography system consisting of two 300 × 7.8 mm BioSep4000 Columns (Phenomenex, Torrance, CA, USA) equilibrated with SEC mobile phase and separated into 80 fractions by a 1200 Series semi-preparative HPLC (Agilent Technologies, Santa Clara, CA, USA) at a flow rate of 0.5 ml/min at 8°C. Fractions 1–55 corresponded to molecular weights 2 MDa to 100 kDa, as determined by the use of common SEC standards thyroglobulin, apoferritin, and bovine serum albumin (BSA) (Sigma-Aldrich) were considered for further analysis. The fractions from the light SILAC lysate served as an internal standard and were separated by SEC independently of the medium/heavy samples. To generate the PCP-SILAC reference mixture, the first 55 fractions of the light SEC-separated samples were pooled together and spiked into each of the corresponding medium/heavy fractions at a volume of 1:0.75 (medium/heavy to light).

### In-solution digestion of PCP-SEC samples

Individual PCP-SILAC samples were prepared using in-solution digestion as previously described (Rogers *et al*, 2010). Briefly, sodium deoxycholate was added to each fraction to a final concentration of 1.0% (v/v) and samples boiled for 5 min. Boiled samples were allowed to cool to RT, then reduced for 60 min with 20 mM dithiothreitol (DTT) at room temperature. Samples were then alkylated for 45 min with 40 mM iodacetamide (IAA) in the dark at room temperature and excess IAA quenched with 40 mM DTT for 20 min. Sequence-grade trypsin (Promega) was added at a ratio of 50:1, and samples were incubated overnight at 37°C. Samples were

acidified to pH < 3 with acetic acid, and the subsequently precipitated deoxycholic acid removed by centrifugation at 16,000 r.c.f. for 10 min. To ensure the removal of particulate matter, peptide digests were further clarified using Unifilter 800 Whatman filter plates (GE Healthcare Life Sciences). The resulting peptide supernatant was purified using self-made Stop-and-go-extraction tips (StageTips) (Rappsilber *et al*, 2007) composed of C18 Empore material (3M) packed in to 200 μl pipette tips. Prior to addition of the peptide solution, StageTips were conditioned with methanol and equilibrated with 0.5% acetic acid (buffer A). Peptide supernatants were loaded onto columns and washed with three bed volumes of buffer A. Peptide samples were eluted with 80% MeCN, 0.5% acetic acid (buffer B) directly into a HPLC autosampler plate, dried down using a vacuum concentrator, and stored at 4°C.

### Preparation of mitochondrial/organelle protein complexes for PCP-BN separation

Mitochondrial/organelle preparations were prepared as previously described (Frezza *et al*, 2007). Briefly, Jurkat cells were harvested, washed with ice-cold PBS, and suspended in ice-cold, freshly prepared isolation buffer (200 mM sucrose, 10 mM Tris–MOPS and 1 mM of EGTA/Tris, pH 7.4 with Complete protease inhibitor cocktail without EDTA). Cells were gently lysed by 200 strokes with a pre-chilled Dounce homogenizer on ice. The lysates were then clarified of nuclei and unbroken cells by centrifugation at 800 r.c.f. for 10 min at 4°C. Mitochondrial/organelle preparations were collected by 7,000 r.c.f. for 10 min at 4°C, washed once with ice-cold isolation buffer, and prepared immediately for BN-PAGE. SILAC samples were prepared in parallel with the same buffers to enhance consistency between preparations.

BN-PAGE separation was performed as outlined by Wittig *et al* (2006). Freshly prepared mitochondrial/organelle preparations were suspended in membrane extraction buffer [4% digitonin stock, 50 mM NaCl, 50 mM imidazole, 2.5 mM 6-aminohexanoic acid, 2 mM EDTA, pH 7.0] and tumbled for 15 min at 4°C to extract protein complexes. Samples were clarified by centrifugation using 16,000 r.c.f. for 10 min at 4°C and the resulting supernatant mixed with a ¼ of the volume of 50% glycerol and ¼ of the volume of 5% Coomassie blue G-250. Membrane complexes were separated using 3–12% gradient NativePAGE™ Bis-Tris gels (Life technologies) with Native running and cathode buffer (Life technologies). Medium and Heavy protein complex extracts were mixed immediately before running of the gels. Gel separation was performed at 4°C, 200 V and gels fixed (10% methanol, 7% acetic acid in Milli-Q water) for 1 h followed by staining overnight with Coomassie blue G-250 (Sigma).

### In-gel digestion of PCP-BN samples

Separated protein complexes were processed as previously described (Shevchenko *et al*, 2006), with minor modifications. Briefly, gel-separated samples were excised into 55 equal gel slices using a OneTouch GridCutter (Gel Company, San Francisco, CA, USA) and destained in a 50:50 solution of 50 mM NH$_4$HCO$_3$: 100% ethanol for 20 min at room temperature with shaking at 750 rpm. Destained bands were then washed with 100% ethanol, vacuum-dried for 20 min, and rehydrated in 10 mM DTT in

50 mM $NH_4HCO_3$. Reduction was carried out for 60 min at 56°C with shaking. The reducing buffer was then removed, and the gel bands washed twice in 100% ethanol for 10 min to ensure the removal of remaining DTT. Reduced ethanol-washed samples were sequentially alkylated with 55 mM iodoacetamide in 50 mM $NH_4HCO_3$ in the dark for 45 min at room temperature. Alkylated samples were then washed twice with 100% ethanol and vacuum-dried. Alkylated samples were then rehydrated with 12 ng/µl trypsin (Promega, Madison WI) in 40 mM $NH_4HCO_3$ at 4°C for 1 h. Excess trypsin was removed; gel pieces were covered in 40 mM $NH_4HCO_3$ and incubated overnight at 37°C. Peptide samples were extracted from gel bands twice using four gel volumes of 30% ethanol 3% acetic acid followed by four gel volumes of 100% ethanol with the supernatant from each extraction pooled. The 0.75 volumes of the resulting pooled peptide extracted mixture from the light reference sample was then added to each fraction from the pooled heavy and medium gel digests. The resulting peptide mixtures were dried down, desalted using C18 StageTips (Rappsilber et al, 2003), and stored on tip at 4°C. Peptides were eluted in buffer B and dried before analysis by LC-MS.

## N-terminal peptide enrichment by TAILS

Protein N-termini were enriched as previously described (Kleifeld et al, 2010, 2011). Labeled cytoplasmic samples were mixed 1:1:1 and precipitated with 20% trichloroacetic acid (TCA) on ice for 10 min. Precipitated protein was collected by centrifugation at 16,000 r.c.f. for 10 min at 4°C, washed with ice-cold acetone, air-dried then re-suspended in 6 M GuHCl, and heated to 99°C for 5 min. Samples were diluted in 3 volumes $dH_20$ and 1 M HEPES pH 8 added to a final concentration of 100 mM. 1 M DTT was added to a final concentration of 5 mM, and samples were incubated 30 min at 60°C. Samples were cooled to room temperature, 100 mM iodoacetamide added to a final concentration of 15 mM, and sample incubated 20 min at room temperature. 2 M $CH_2O$ was added to each sample to a final concentration of 40 mM followed by 1 M $NaBH_3CN$ to a final concentration of 20 mM. The pH was adjusted to between 6 and 7 and samples incubated overnight at 37°C. Fresh $CH_2O$ and $NaBH_3CN$ were added, and samples incubated for an additional 1 h at 37°C to ensure complete amine labeling. Excess $CH_2O$ was quenched by adding 1 M $NH_4HCO_3$ to a final concentration of 100 mM, the pH adjusted to between 6 and 7, and samples incubated for 1 h at 37°C. Samples were precipitated with 20% TCA; 1 volume TCA per 4 sample volumes was added and incubated 10 min at 4°C. Proteins were pelleted at 16,000 r.c.f for 5 min and washed 3× with 500 µl ice-cold acetone. Proteins were re-suspended in ice-cold 100 mM NaOH and then adjusted to 1 mg/ml in 50 mM HEPES pH 8. Trypsin was added at a protease:protein ratio of 1:100 and incubated > 18 h at 37°C. HPG-ALDII (FlintBox, http://flintbox.com/public/project/1948/) was added at a HPG-ALDII:peptide ratio of 5:1 and coupling commenced by the addition of 1 M $NaBH_3CN$ to a final concentration of 20 mM. The pH was adjusted to between 6 and 7, and samples were incubated overnight at 37°C. 1 M $NH_4HCO_3$ was added to a final concentration of 100 mM to quench the HPG-ALDII, and samples were incubated at 37°C for 30 min. Unbound peptides were collected by using a 10K spin column (Amicon

Ultra-0.5 Centrifugal Filter Unit with Ultracel-10 membrane), cleaned up using C18 StageTips (Rappsilber et al, 2003), and stored 4°C.

## Liquid chromatography and mass spectrometry analysis

Prior to LC-MS analysis, samples were re-suspended in 15 µl buffer A. LC-MS was performed on either an Agilent 1290 Series HPLC (Agilent Technologies, Mississauga, ON) coupled to LTQ-Orbitrap Velos (Thermo Scientific, San Jose, CA) for the establishment of Fas apoptosis time points and development of the PCP-BN-PAGE method or an EASY-nLC1000 system coupled to a Q-Exactive for biological replicates of the quantitative studies. LC-MS was accomplished using a two column system in which samples were concentrated prior to separation on a 2-cm-long, 100-µm-inner-diameter fused silica trap column containing 5-µm Aqua C18 beads (Phenomenex) and then separated using an in-house-packed C18 analytical 75-µm inner diameter, 360-µm outer diameter column composed of 35 cm ReproSil-Pur C18 AQ 1.9 µm (Dr. Maisch, Ammerbuch-Entringen, Germany) column for the EASY-nLC1000 system, or a 20 cm ReproSil-Pur C18 AQ 3 µm for the Agilent 1290 Series HPLC. Samples were concentrated onto the trap for 10 min using 100% buffer A at 5 µl/min after which the gradient was altered from 100% buffer A to 40% buffer B over 180 min at 250 nl/min with the eluting peptides infused directly into the mass spectrometers via nESI. Both instruments were operated in a data-dependent manner using Xcalibur v2.2 (Thermo Scientific). For samples analyzed on the LTQ-Velos, one full precursor scan in the Orbitrap (resolution 60,000; 350–1,600 Th, AGC target of $1 \times 10^6$) was followed by the selection of the top ten most intense, multiply charged ions above 1,000 counts for collision-induced dissociation (normalized collision energy 35, activation Q, 0.25; activation time, 10 ms, AGC of $4 \times 10^4$) with 30-s dynamic exclusion enabled. For samples analyzed on the Q-Exactive, one full precursor scan (resolution 70,000; 350–2,000 m/z, AGC target of $3 \times 10^6$) was followed by 10 data-dependent HCD MS-MS events (resolution 17.5 k AGC target of $1 \times 10^6$ with a maximum injection time of 60 ms, NCE 28 with 20% stepping) with 25-s dynamic exclusion enabled. TAILS samples were analyzed on a Q-Exactive with the parameters optimized for greater spectral quality with one full precursor scan (resolution 70,000; 350–2,000 m/z, AGC target of $3 \times 10^6$) followed by ten data-dependent HCD MS-MS events (resolution 35 k AGC target of $1 \times 10^6$ with a maximum injection time of 110 ms, NCE 28 with 35% stepping).

## Data analysis

MaxQuant (v1.4.1.2 or 1.5.0.0) (Cox & Mann, 2008) was used for identification and quantification of the resulting experiments, with the resulting cytoplasmic and membrane biological replicates searched together to ensure a global false discovery rate of < 1% in accordance with the work of Schaab et al (2012). Database searching was carried out against the UniProt/Swiss-Prot human database (downloaded 24/10/2013) (84,843 entries) with the following search parameters: carbamidomethylation of cysteine as a fixed modification; oxidation of methionine, acetylation of protein N-termini, trypsin/P cleavage with a maximum of two missed cleavages. A multiplicity of three was used, with each multiplicity

denoting one of the SILAC amino acid combinations (light, medium, and heavy, respectively). The precursor mass tolerance was set to 6 parts-per-million (ppm) and MS/MS tolerance 0.5 Da for LTQ-Velos data or 10 ppm for Q-Exactive data, with a maximum false discovery rate of 1.0% set for protein identifications. To enhance the identification of peptides between fractions and replicates, the Match between Runs option was enabled with a precursor match window set to 2 min and an alignment window of 10 min. The resulting protein group output was processed within the Perseus (v1.4.0.6) analysis environment to remove reverse matches and common proteins contaminates prior to analysis with MATLAB R2012a (http://www.mathworks.com). Enrichment analysis was performed within Perseus, utilizing Cellular Component GO terms and the CORUM database (Ruepp et al, 2010).

Terminal amine isotopic labeling of substrates data were converted to MGF using MSConvert (Chambers et al, 2012) and then searched using MASCOT 2.5. Data files were search using semi-ArgC specificity, carbamidomethylation of cysteine as a fixed modification and the following variable modifications: Acetyl (N-term), Dimethyl (K), Dimethyl (N-term), Gln->pyro-Glu (N-term Q), Label:13C(6) (R), Label:13C(6)15N(2) + Dimethyl (K), Label:13C(6) 15N(4) (R), Label:2H(4) + Dimethyl (K), Oxidation (M). A precursor and product tolerance of 20 ppm was used against the UniProt/ Swiss-Prot human database (84,843 entries). Quantitation of resulting TAILS peptides was accomplished using Skyline v3.1 utilizing the MS1 full scan features (Schilling et al, 2012). Four quantitation channels were created as follows; light, corresponding to no modified lysine or arginine residues; Light_dimethyl, corresponding to dimethylated lysine and unlabeled arginine; medium, corresponding to dimethylated lysine-4 and arginine-6; and heavy, corresponding to dimethylated lysine-8 and arginine-10. The results from Mascot searches were used to generate spectral libraries that were then used to extract quantitative values for the four possible channels. The resulting results were compiled and filtered using MATLAB to generate a complete list of the quantitative values for all peptides which pass a 1% FDR. For statistical analysis, multiple hypothesis corrections were implemented using a Benjamini–Hochberg correction threshold below 0.02 for enrichments analysis, 0.2 for TAILS fold change analysis, and Bonferroni correction below 0.05 for PCP-SILAC fold change analysis. Cleavage motif analysis of dimethylated N-termini peptide observed to change in response to treatments were accomplished using pLogo (O'Shea et al, 2013).

### MATLAB-based analysis

All experiments were processed using the bioinformatics approach outlined within Scott et al (2015). In addition to the Gaus.m, Alignment.m, Comparison.m and PPI.m scripts described previously, additional scripts membrane.m, peptide_mapping.m, and cluster.m were created to further enhance the analysis and visualization of data. The precision of protein interaction networks generated during analysis was calculated at two levels, at the local level; corresponding to all interactions within an isotopologue channel of a single experiment, and at the global level; corresponding to the combined precision of all interaction across all isotopologue channels and replicates. The desired precisions of 70, 60, and 50% were generated by optimization of the parameters required to reach these precisions threshold at the local level

within a given isotopologue channel and then combining the determined interactions into a single networks. Membrane.m enables GO terms possessed by assigned interactions to be assessed based on the percentage of membrane-associated GO terms and matching terms. Peptide_mapping.m enables the analysis and visualization of peptide information for individual protein groups within a PCP experiment, similarly to a previous report (Stoehr et al, 2013). Clustering was accomplished using a Markov clustering approach (Enright et al, 2002; Guruharsha et al, 2011; Babu et al, 2012). Peptide quantitative information is utilized to assess evenness and diversity using a Shannon index (Shannon & Weaver, 1949; Bent & Forney, 2008).

### Microscopy and co-localization analysis

Cells for microscopy were subject to the same growth and treatment/control conditions as the cells destined for proteomic analysis. Following treatment, the cells were moved to tubes, centrifuged for 5 min at 200 RCF, and the culture medium removed. The cells were re-suspended in PBS containing 200 nM MitoTracker Orange (ThermoFisher Scientific, M7510) for 30 min at 37°C to label mitochondria. Following this, $1.5 \times 10^5$ cells in a 0.5 ml volume were centrifuged onto 12-mm-diameter, thickness 1.5 glass coverslips for staining and microscopy. The cells were stained with mouse monoclonal anti-CDC24 (Santa Cruz Biotechnology, sc-8401) at 1:50, and rabbit purified polyclonal anti-IQGAP1 (BD Biosciences, HPA014055) at 1:100 for 30 min at room temperature. Goat anti-mouse Alexa Fluor 594 and goat anti-rabbit Alexa Fluor 488 antibodies were applied at a 1:500 dilution, together with 1 μM TO-PRO-3 iodide to visualize nuclei (ThermoFisher Scientific, A-11032, A-11034, and T3605, respectively). Microscopy was performed on a Leica SP5 using a pinhole of 1 Airy unit, and image data were collected at a depth of 12-bit per channel. Excitation wavelengths for Alexa Fluor 488, MitoTracker Orange, Alexa Fluor 594, and TO-PRO-3 were 488-, 543-, 594-, and 633-nm laser lines, and emission windows on the spectral scanning unit were set to 493–540, 550–590, 600–645, and 645–740 nm, respectively. Images were processed and analyzed using ImageJ (Schindelin et al, 2015). Noise was reduced by applying Gaussian Blur 3D with a value of 1 in all dimensions, and Spearman's rank correlation values were determined as the measure of colocalization for each cell as a hand-selected region-of-interest (Dunn et al, 2011). Statistical significance of changes in colocalization upon stimulation of apoptosis was assessed by performing a Mann–Whitney U-test with a 95% confidence interval for treated cells showing mitochondrial or nuclear morphological changes associated with apoptosis vs. untreated cells that showed no such morphological signs.

### Granzyme B activity assays

Cells were grown and treated as for the proteomics experiments. Following 4-h treatments, cells were collected by centrifugation, washed with PBS, and re-suspended in 0.1 ml of the assay buffer (50 mM HEPES pH 7.4, 0.05% CHAPS, 5 mM DTT) lysed by 20 passages through the 26-gauge syringe needle and spun down (16,000 r.c.f., 10 min at 4°C). One half of each supernatant was incubated with 50 μM Compound 20 (Willoughby et al, 2002; provided in kind by Dr. David Granville, UBC) for 30 min on ice to

specifically inhibit granzyme B activity. The reactions were initiated by the addition of the substrate Ac-IEPD-pNA (20 μM final; Grujic *et al*, 2005). Release of pNA was monitored by the absorbance at 405 nm and using the SpectraMax plate reader (Molecular Devices) at 37°C. The calculated rates were normalized per total protein (as measured by Bradford assays in the same lysates) with experiments preformed in biological triplicate.

### Data availability

All mass spectrometry data have been deposited to the ProteomeXchange Consortium (Vizcaino *et al*, 2014) via the PRIDE partner repository with the dataset identifier PXD002892. A complete list of all data used within this study is provided within Table EV24. All the scripts described, together with representative test datasets, are available as Code EV1.

**Expanded View** for this article is available online.

### Acknowledgements

CIHR Open Operating Grant (MOP-77688) to LJF provided financial support for this work. The Canada Foundation of Innovation, the British Columbia Knowledge Development Fund, and the BC Proteomics Network supported the mass spectrometry infrastructure used within the work. NES was supported by a Michael Smith Foundation Post-doctoral Fellowship (award # 5363) and a National Health and Medical Research Council of Australia (NHMRC) Overseas (Biomedical) Fellow (APP1037373). CMO is a Canada Research Chair in Proteinase Proteomics and Systems Biology. We would like to thank Dr. David Granville (UBC) for a kind gift of Compound 20.

### Author contributions

NES and LJF conceived and designed the experiments. NES, LDR, NFB, and AP performed the experimental work. NES and NF performed the data analysis. NES, NFB, AP, CMO, and LJF wrote and edited the manuscript.

### Conflict of interest

The authors declare that they have no conflict of interest.

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
