## [Review Process File · Molecular Systems Biology]

Interactome disassembly during apoptosis occurs independent of caspase cleavage

Nichollas E. Scott, Lindsay D. Rogers, Anna Prudova, Nat F. Brown, Nikolaus Fortelny, Christopher M. Overall and Leonard J. Foster

Corresponding author: Leonard Foster, University of British Columbia

Review timeline:

Submission date:	16 May 2016
Editorial Decision:	30 June 2016
Revision received:	24 October 2016
Editorial Decision:	28 November 2016
Revision received:	05 December 2016
Accepted:	12 December 2016

Editor: Maria Polychronidou

Transaction Report:

1st Editorial Decision

30 June 2016

Thank you again for submitting your work to Molecular Systems Biology. We have now heard back from two of the three referees who agreed to evaluate your study. As you will see below, the reviewers acknowledge the extensive and high-quality datasets. However, they raise a number of concerns, which should be carefully addressed in a revision of the manuscript.

Without repeating all the points listed below, one of the more fundamental issues raised by both referees is that further analyses providing some level of functional insight into the role of protein complex remodeling in apoptosis would significantly enhance the overall impact of the study.

REFeree REPORTS

Reviewer #2:

Summary

This paper presents a proteomic look at Fas-mediated apoptosis. Specifically, the authors investigated protein correlation networks and their TAILS technology to look into the relationship between apoptotic proteolysis and dissolution of protein complexes. They generated a large dataset of complexes using a new method to isolate both membrane bound and cytosolic proteins. This was then compared to N-termini generated from untreated, fas-mediated apoptotic samples and caspase inhibited samples. They discovered a lot of interactions, and correlated many of them with known complexes while adding some new interactions. The major finding of their paper comes with their connection of these complexes during apoptosis, indicating that there is protein complex dissolution

occurring before caspase cleavage, potentially allowing for caspase cleavage.

General remarks

The study is well-designed and rigorously performed. The design and data analysis is done carefully and all the reasoning is well-explained. While multiple studies have investigated protein complexes and apoptosis, this current study is a much larger and provides an unbiased dataset. Additionally, while the study was primarily focused on apoptotic proteolysis, the datasets will be a useful reference for protein interactions and N-termini. They also demonstrated an improved technique for membrane interaction methods. Their data and logic do support their main conclusion that many apoptotic interactions are altered independently of caspase cleavages. I recommend publication after addressing the points below.

Major points

However, while their paper presents a very strong case, they need to address some issues. First, a key observation is that there are changes in the interactome that precede the bulk of caspase cleavage events. However, only a small fraction (~2%) of the interactome changes in apoptosis (~300 of ~18,000 observed). These 300 are enriched for known caspase sites but data is presented that these are not extensively cleaved at the 4hr time point after Fas treatment. Given that these are in complexes is it possible only one cleavage in the complex would be sufficient to disrupt it. For example, if the target is a tetramer, cleavage of only 1 subunit may be sufficient for disruption, or if a larger complex the possibility is even stronger. The authors should comment on this possibility and unless they have data to suggest the contrary. Secondly, they assume these complexes must be "key" to driving the apoptotic process, but no data is presented showing their essentiality to the process. As such they may be early events but may not actually be drivers. They should clarify what they mean by key complexes and if they do not provide evidence for essentiality they should state that clearly. Finally, their conclusions leave us wondering what may be causing the disassembly during the early stages of apoptosis. Is it a post-translational or another protein-protein interaction. What would they speculate as causing the disassembly if not proteolysis?

During this study, they discover some interesting N-termini in the Z-vad FMK treated cells. Specifically, proteolytic activity and non-caspase inhibitor sensitive aspartic cleavages they attribute to granzymes A and B. Choosing specific granzyme targets of interest identified here to follow cleavage in granzyme inhibited with and without Z-vad FMK in fas-mediated apoptosis by mass spectrometry or western blot would greatly support this novel point.

They show in Figure 5E that there is little correlation between processing and disassembly. Is there any key distinguishing feature that may group the quadrants by function, abundance, cut site location or known caspase/apoptotic target, etc?

Minor points

Figure 1 - It is not immediately clear how the mass gradient fits with the plots as the x-axis label is shared and at the bottom. It looks like C/D/E are labeled incorrectly.

Figure EV1- No A or B on figure itself.

Figure EV5- Differing titles on the logos makes it a little confusing to compare them.

Figures- All figures with venn diagrams seem to have ghost lines visible on the high resolution digital version.

Reviewer #3:

The manuscript by Scott et al. describes the application of two different proteomic methods - protein correlation profiling (PCP) and terminal amine labeling of substrates (TAILS) - to study apoptosis. PCP can be used to reconstruct protein-protein interaction networks while TAILS allows systematic analysis of protein processing. Thus, these two methods in combination can be used to investigate potential links between both processes during apoptosis. The key finding is that proteolytic

processing and complex disassembly do not seem to be generally linked. The authors conclude that complex disassembly precedes proteolytic processing.

This manuscript presents an impressive amount of data. In fact, there is so much data (32 supplemental tables) that I simply cannot look into all of the details. Since the Foster and Overall lab have shown before that they can master the PCP and TAILS technology I assume the data presented here is also of high quality without being able to check all the details. My main critique is that the study is overall rather descriptive with little validation of the key findings. Moreover, it is difficult to understand the experiments and interpretation of the data since the Results section is not very clear. I therefore think this paper requires additional validation experiments and should be re-written with more focus on clarity before being acceptable for publication in MSB.

Specific points:

- 1.) It is quite difficult to read and understand this manuscript, especially the Results section. I had a hard time working my way through it. This section should be re-written with more focus on overall clarity rather than on all of the dozens of supplemental tables, figures etc. The Discussion section is much better in this respect. Some of the figures should also be to clarify what they actually show. The key Figure EV1 that shows the experimental design was particularly confusing to me.
- 2.) Overall, this is a rather descriptive story: an impressive amount of data with not so many biological insights. The main conclusion - poor correlation between complex disassembly and protein processing - is interesting. However, it is not supported by follow-up experiments. Figure 6 shows that several known caspase targets are not processed 4 h post treatment, but the change in their interactomes of these proteins at his time point is not validated. The observation that granzymes seem to be involved in proteolytic cleavages is also intriguing but this is also not validated or followed up functionally.
- 3.) Figure EV1 is not very clear. Since I am familiar with the technologies I can actually guess what most of the items are supposed to show, but this will be much more difficult for other readers. Figure S1 is much clearer. For example, where does the workflow start and end (I guess in the middle, but it took me a while to realize this)? What is the meaning of the depicted machines, one at the top left and one at the bottom left (I guess these are mass spectrometers)? What does the bar chart in the top right indicate? What is this tube (I guess an SEC column)? What is this blue, red and green-colored rectangle (I guess a BN gel)? In general, how are the different parts of the figure related to each other?
- 4.) Page 6: "Thus, we applied PCP-SILAC to analyze membrane protein complexes resolved by BN-PAGE [...] from a mitochondrial membrane preparation". This sentence implies that only BN-PAGE was used for fractionation. In contrast, Figure EV1 indicates that SEC was also used. Also, was BN only done on mitochondrial membrane preparations? This should be better explained in the main text.
- 5.) Page 6: "Initially we used the medium and heavy SILAC channels to compare two technical replicates for reproducibility". This is inconsistent with Fig. EV1. In the figure, light and medium-heavy cells are replicates while heavy-cells were treated with anti-FAS.
- 6.) Page 6: "Reproducibility of quantitation [...] was very high [...], proving that this approach can accurately quantify interactome changes." The problem here is that samples from both replicates were combined with the common reference before fractionation. Therefore, we do not know how reproducible the fractionation really is. The reproducibility will be considerably lower when two completely independent fractionation experiments are compared to each other. This should at least be acknowledged.
- 7.) Fig. 1 B: I don't understand this pie chart. Are only GO terms which are shared between both proteins in an interacting pair displayed? At first sight it looks very impressive that only 3% of interactions don't share a GO CC term. However, this critically depends on the level of the term in the GO hierarchy. For example, all proteins with available GO CC annotation will of course share the top level term "cellular component". I also don't understand how the authors treated proteins that don't have annotated GO terms.

8.) Page 8, "The segregation of mitochondrial and cytosolic complexes was nearly complete, with less than 1% of the total interactions being shared". This might simply result from an overall low coverage (i.e., high false negative rate) of interactions.

9.) I would have expected a good coverage of the mitochondrial respiratory chain complexes in the dataset. However, looking at a few selected examples, the coverage does not seem to be good. More importantly, sometimes proteins that are part of different complexes are reported to interact. The only interaction partner in Table S6 of P00156 (Cytochrome B, a central protein in complex III) is P56181-2, which is part of complex I. COXII (P00403) only interacts with COX6C (P09669) and none of the other members of complex iv.

10.) I don't fully understand the logic behind Table S6. The first column header is "unique interactions" but actually several of these occur multiple times. For example, "P00403_P09669" is listed three times, all detected in both replicates. I don't understand why this is. What causes this redundancy?

11.) The first paragraph of the introduction is very general with little relevance to the investigated question (that is, changes in the interactome during apoptosis). I think it should be shortened.

12.) Page 3, line 20, should be "differ in the role OF mitochondria"

13.) Fig. S3, legend: "Gel slices are generated and in- gel digestion preformed.", should be performed

14.) It would be very helpful and increase readability to number the main text figures.

15.) Fig. 1 A: "Gaussians features" should be "Gaussian features"

16.) Page 9: "This could occur in one of two ways: either proteolysis drove the rearrangement of the complexes or rearrangement has led to their subsequent proteolysis". There is of course also a third possibility: Proteolysis and complex rearrangement could be two rather independent processes that are both induced by apoptosis. The significant overlap with Degradase might have other reasons.

Reviewer #2:

Summary

This paper presents a proteomic look at Fas-mediated apoptosis. Specifically, the authors investigated protein correlation networks and their TAILS technology to look into the relationship between apoptotic proteolysis and dissolution of protein complexes. They generated a large dataset of complexes using a new method to isolate both membrane bound and cytosolic proteins. This was then compared to N-termini generated from untreated, fas-mediated apoptotic samples and caspase inhibited samples. They discovered a lot of interactions, and correlated many of them with known complexes while adding some new interactions. The major finding of their paper comes with their connection of these complexes during apoptosis, indicating that there is protein complex dissolution occurring before caspase cleavage, potentially allowing for caspase cleavage.

General remarks

The study is well-designed and rigorously performed. The design and data analysis is done carefully and all the reasoning is well-explained. While multiple studies have investigated protein complexes and apoptosis, this current study is a much larger and provides an unbiased dataset. Additionally, while the study was primarily focused on apoptotic proteolysis, the datasets will be a useful reference for protein interactions and N-termini. They also demonstrated an improved technique for membrane interaction methods. Their data and logic do support their main conclusion that many apoptotic interactions are altered independently of caspase cleavages. I recommend publication after addressing the points below.

Major points

1. “However, while their paper presents a very strong case, they need to address some issues. First, a key observation is that there are changes in the interactome that precede the bulk of caspase cleavage events. However, only a small fraction (~2%) of the interactome changes in apoptosis (~300 of ~18,000 observed). These 300 are enriched for known caspase sites but data is presented that these are not extensively cleaved at the 4hr time point after Fas treatment. Given that these are in complexes is it possible only one cleavage in the complex would be sufficient to disrupt it. For example, if the target is a tetramer, cleavage of only 1

subunit may be sufficient for disruption, or if a larger complex the possibility is even stronger.

The authors should comment on this possibility and unless they have data to suggest the contrary.”

The reviewer highlights an interesting possibility that a single cut of any member of a given complex may lead to disassembly of the complex. We agree that this concept should be discussed in the manuscript and have added the following paragraph on page 25:

“Although our and Stoehr *et al* observations support minimal proteolysis during the early stages of apoptosis, these data do not provide complete details on its functional consequences: it is currently unknown whether complex disassembly can be triggered by cleavage of any individual complex member or whether it requires a critical mass of cuts in several complex members. These dynamics would be highly unique for each specific complex and defined by its features, such as number of subunits, affinities and kinetics of their interaction, specific location of a cleavage(s) and its penetration within the total pool of that specific protein subunit.”

2. “Secondly, they assume these complexes must be “key” to driving the apoptotic process, but no data is presented showing their essentiality to the process. As such they may be early events but may not actually be drivers. They should clarify what they mean by key complexes and if they do not provide evidence for essentiality they should state that clearly.”

We thank the reviewer for pointing out a potential source of ambiguity. To clarify, there is evidence in literature that some of the complexes we observe to disassemble during apoptosis are key to driving the process (Li *et al*, 1998; Sakahira *et al*, 1998). In addition, within the manuscript we highlight that the complexes that are lost are enriched for known caspase targets, some of which were shown (Li *et al*, 1998) to be key drivers, accelerants and initiators of apoptosis. The initial use of the word “key” to describe these complexes within the abstract sought to draw the attention to this link. However, to avoid any confusion we have now changed the word “key” to “select”.

To further clarify this within the manuscript we have added the following statement on page 24: “As the disassembly of these complexes occurs early during apoptosis it is tempting to speculate that the loss of these complexes drives apoptosis akin to cleavage of caspase targets such as ICAD (Sakahira *et al*, 1998) and BID (Li *et al*, 1998). However, the essentiality of the loss of these complexes to the initiation of apoptosis has yet to be determined.”

3. Finally, their conclusions leave us wondering what may be causing the disassembly during the early stages of apoptosis. Is it a post-translational or another protein-protein interaction. What would they speculate as causing the disassembly if not proteolysis?

The causes of the observed interactome changes are most likely multifaceted yet the scope and nature of the changes suggest a mechanism, which affects multiple complexes simultaneously, consistent with a post-translational modification. Another possibility is an expected change in localization and concentrations of the subunits upon mitochondrial damage. To address this comment we have added the following statement to the discussion, page 26: “Although the cause of these dramatic interactome changes are unknown, the scope of complex remodelling suggests that the mediator acts in a rapid, pleiotropic manner consistent with initiation by a protein modification. Intriguingly, modifications such as phosphorylation and glycosylation have both been shown to augment apoptosis (Dix *et al*, 2012; Zhu *et al*, 2001), yet the potential connection of these protein modifications to interactome rearrangement remains to be tested.”

4. During this study, they discover some interesting N-termini in the Z-vad FMK treated cells. Specifically, proteolytic activity and non-caspase inhibitor sensitive aspartic cleavages they attribute to granzymes A and B. Choosing specific granzyme targets of interest identified here to follow cleavage in granzyme inhibited with and without Z-vad FMK in fas-mediated apoptosis by mass spectrometry or western blot would greatly support this novel point.

In accordance with the reviewer's suggestion we have taken a multi-pronged approach to investigate the potential origins of these unusual N-termini. After attempting to confirm the cleavage of multiple targets we found that similar to the caspase targets at 4hours post Fas-treatment little evidence for large proteins changes could be observed as can be seen below in the examples T-complex protein 1 subunit theta (P50990) and Elongation factor 1-alpha 1 (P68104), which are cleaved C-terminal to aspartic acid in a Z-vad FMK insensitive manner.

Reviewer comment figure 1: Western analysis of Z-vad FMK insensitive cleavage substrates T-complex protein 1 subunit theta (P50990) and Elongation factor 1-alpha 1 (P68104). Minimal evidence of degradation can be observed.

As this observation is consistent with the concept that at the early stages of apoptosis the effect on the protein level can be subtle, consistent with the work of Stoehr *et al*, we next attempted to confirm enzymatic activity directly. Utilizing the Granzyme B specific inhibitor Compound 20 we noted that no decrease in proteolysis was observed in Jurkat cells treated with this inhibitor (Appendix Figure S11B). This result suggested Granzyme B was not responsible for the observed N-termini and upon probing directly for Granzyme B (Appendix Figure S11A) we noted that within our Jurkat cell line granzyme B is absent. This observation is consistent with previous reports that unless Jurkat cells are treated to force differentiation to a more cytotoxic T cell like state granzyme B can be undetectable. These findings rule out Granzyme B as the possible origins of these termini in favour of either a Z-vad FMK insensitive caspase or a currently unknown proteases involved in the early stages of apoptosis.

5. They show in Figure 5E that there is little correlation between processing and disassembly. Is there any key distinguishing feature that may group the quadrants by function, abundance, cut site location or known caspase/apoptotic target, etc?

Based on our analyses we have been unable to find any distinguishing feature that links processing and disassembly. This said the lack of structural information for the majority of proteins have limited our ability to find structural similarities or features, which may link these phenomena.

Minor points

1. Figure 1 - It is not immediately clear how the mass gradient fits with the plots as the x-axis label is shared and at the bottom. It looks like C/D/E are labeled incorrectly.

We thank the reviewer for pointing out the mislabeled panels C/D and E, this mistake has been fixed. To improve the utility and the clarity of the figure, the mass gradient has been moved and an additional explanation added to the figure legend.

2. Figure EV1- No A or B on figure itself.

In accordance with reviewers 2 and 3 concerns, this figure has been separated into two separate panels labeled A and B.

3. Figure EV5- Differing titles on the logos makes it a little confusing to compare them.

We thank the reviewer for pointing out the inconsistency within the titles, which have been changed to remove ambiguity.

4. Figures- All figures with venn diagrams seem to have ghost lines visible on the high resolution digital version.

The ghost lines within Venn Diagrams have been removed, we thank the reviewer for bringing this to our attention.

Reviewer #3:

1. The manuscript by Scott et al. describes the application of two different proteomic methods - protein correlation profiling (PCP) and terminal amine labeling of substrates (TAILS) - to study apoptosis. PCP can be used to reconstruct protein-protein interaction networks while TAILS allows systematic analysis of protein processing. Thus, these two methods in combination can be used to investigate potential links between both processes during apoptosis. The key finding is that proteolytic processing and complex disassembly do not seem to be generally linked. The authors conclude that complex disassembly precedes proteolytic processing.

This manuscript presents an impressive amount of data. In fact, there is so much data (32 supplemental tables) that I simply cannot look into all of the details. Since the Foster and Overall lab have shown before that they can master the PCP and TAILS technology I assume the data presented here is also of high quality without being able to check all the details. My main critique is that the study is overall rather descriptive with little validation of the key findings.

We acknowledge the reviewer's comment on the descriptive nature of the work but would like to point out that this is the first study ever exploring the interplay between PTMs (i.e. proteolysis) and protein-protein interactions at a proteome-wide level. Thus, as a global proof-of-principle analysis, this work is somewhat descriptive by design. However, we have now supported the findings of this work by the addition of confocal microscopy, enzymatic activity assays and additional western analyses.

Moreover, it is difficult to understand the experiments and interpretation of the data since the Results section is not very clear. I therefore think this paper requires additional validation experiments and should be re-written with more focus on clarity before being acceptable for publication in MSB.

To improve clarity, we have re-written multiple sections of the manuscript and provided additional experimental evidence.

Specific points:

1.) It is quite difficult to read and understand this manuscript, especially the Results section. I had a hard time working my way through it. This section should be re-written with more focus on overall clarity rather than on all of the dozens of supplemental tables, figures etc. The Discussion section is much better in this respect. Some of the figures should also be to clarify what they actually show. The key Figure EV1 that shows the experimental design was particularly confusing to me.

We agree that figure EV1 was not as clear as it should have been and changed this to enhance clarity. As requested, we have re-written multiple sections of the manuscript and provided additional experimental evidence.

2.) Overall, this is a rather descriptive story: an impressive amount of data with not so many biological insights. The main conclusion - poor correlation between complex disassembly and protein processing - is interesting. However, it is not supported by follow-up experiments. Figure 6 shows that several known caspase targets are not processed 4 h post treatment, but the change in their interactomes of these proteins at this time point is not validated. The observation that granzymes seem to be involved in proteolytic cleavages is also intriguing but this is also not validated or followed up functionally.

To improve the manuscript, we have further validated the observation of this study with complementary approaches. Previous studies have shown the progression to committed cellular destruction by apoptosis is rapid yet cells within a population progress to this end-point at different rate. In light of this we reasoned the examination of proteins on a population level, using proteomics or western blotting may be masking dramatic effect on proteins which only occur when cells are examined at the single cell level. To address this, we have undertaken confocal microscopy examination of CDC42 and its binding partner IQGAP1 and how the co-localization of these proteins change in response to Fas-mediated apoptosis. The finding of these experiments show that under non-treated conditions CDC42 and IQGAP1 co-localize, consistent with these proteins interacting yet in response to Fas treatment the signal for CDC42 is rapidly lost within cells committed to apoptosis (Figure 6B and C).

Furthermore, we have investigated and excluded granzyme B involvement in early apoptosis in Jurkat cells by western blotting and activity assays. From these experiments we have found that the Z-vad FMK insensitive Asp cleavage products are not due to Granzyme B activity but rather a product of another Z-vad FMK protease, currently incognito.

3.) Figure EV1 is not very clear. Since I am familiar with the technologies I can actually guess what most of the items are supposed to show, but this will be much more difficult for other readers. Figure S1 is much clearer. For example, where does the workflow start and end (I guess in the middle, but it took me a while to realize this)? What is the meaning of the depicted machines, one at the top left and one at the bottom left (I guess these are mass spectrometers)? What does the bar chart in the top right indicate? What is this tube (I guess an SEC column)? What is this blue, red and green-colored rectangle (I guess a BN gel)? In general, how are the different parts of the figure related to each other?

In accordance with the reviewer's suggestions figure EV1 has been modified to improve clarity and separated into two separate panels A and B denoting the two separate workflows. The figure legend has been expanded to further explain both workflows.

4.) Page 6: "Thus, we applied PCP-SILAC to analyze membrane protein complexes resolved by BN-PAGE [...] from a mitochondrial membrane preparation". This sentence implies that only BN-PAGE was used for fractionation. In contrast, Figure EV1 indicates that SEC was also used. Also, was BN only done on mitochondrial membrane preparations? This should be better explained in the main text.

In accordance with the reviewer's suggestions we have modified the outlined rationale to state "Studies of interactome-wide changes are rare but they are non-existent for organelle or membrane interactomes. As PCP-SILAC enables the measurement of cytosolic interactome responses (Kristensen et al, 2012), we reasoned that using a membrane-compatible separation method should allow the measurement of organelle/membrane interactome dynamics. SEC provides a robust workflow for the separation of cytoplasmic complexes however it is not compatible with membrane complexes as they are extremely sensitive to separation conditions (Babu et al, 2012; Drew et al, 2008). Thus, to analyze membrane protein complexes we utilized PCP-SILAC and BN-PAGE, a separation approach known to be broadly applicable to membrane complexes (Wittig et al, 2006) (BN-PCP-SILAC, Appendix Figure S1)."

5.) Page 6: "Initially we used the medium and heavy SILAC channels to compare two technical replicates for reproducibility". This is inconsistent with Fig. EV1. In the figure, light and medium-heavy cells are replicates while heavy-cells were treated with anti-FAS.

The initial optimization experiments to assess the variability of BN-PAGE experiments were conducted as outlined within manuscript. We understand the confusion of the reviewer as we refer to EV1 where supplementary Figure S1 provides a clearer visualization of how this experiment was undertaken. Although we refer to Figure S1 at the beginning of this paragraph we have modified Figure S1 to explicitly state the initial experiments were conducted with two populations of cells, which were both untreated. Furthermore, we have added a reference to Figure S1 within the originally confusing sentence which now states "Initially we used the medium and heavy SILAC channels to compare two technical replicates for reproducibility (Appendix Figure S1)".

6.) Page 6: "Reproducibility of quantitation [...] was very high [...], proving that this approach can accurately quantify interactome changes." The problem here is that samples from both replicates were combined with the common reference before fractionation. Therefore, we do not know how reproducible the fractionation really is. The reproducibility will be considerably lower when two completely independent fractionation experiments are compared to each other. This should at least be acknowledged.

As the isolation of membrane complexes from each sample is done independently the reproducibility referred to within this statement is the ability to accurately isolate and then compare membrane complexes between samples. As outlined within Kristensen *et al* 2012 the key strength of utilizing SILAC-based quantitation for PCP is to enable the combination of samples prior to separation. This enables accurate assessment of interactome changes as both interactome undergo identical separation conditions. The reviewer is correct in that reproducibility is lower between independent experiments. However, using our computational pipeline we are able to re-align PCP experiments thus reducing the errors which can arise from differences in chromatographic separation. To address the reviewer's concerns we have added the following statement within the results "Importantly, the utilization of our bioinformatics pipeline (Scott et al, 2015) enabled the re-alignment and

quantitation of features across biological replicates overcoming variability resulting from independent fractionation experiments.”

7.) Fig. 1 B: I don't understand this pie chart. Are only GO terms which are shared between both proteins in an interacting pair displayed? At first sight it looks very impressive that only 3% of interactions don't share a GO CC term. However, this critically depends on the level of the term in the GO hierarchy. For example, all proteins with available GO CC annotation will of course share the top level term "cellular component". I also don't understand how the authors treated proteins that don't have annotated GO terms.

Figure 1B was generated by comparing all assigned GO CC terms for proteins within interaction pairs with the GO provided by Uniprot (downloaded 04-07-2014). Only GO terms below the “cellular component” are included within this analysis, thus term cellular component is excluded. As shown in table EV7, the overlap between terms can be assessed in multiple ways: by directly comparing identical terms assigned to both proteins in a interaction pair or examining if any of the GO CC terms for each protein of an interaction pair contains the word membrane. In cases where no GO terms are assigned for proteins no comparison can be made and these cases are excluded from analysis. For transparency we have provided the data, which was utilized to generate figure 1B as table EV7. This is stated within the manuscript whenever Figure 1B is referred to.

8.) Page 8, "The segregation of mitochondrial and cytosolic complexes was nearly complete, with less than 1% of the total interactions being shared". This might simply result from an overall low coverage (i.e., high false negative rate) of interactions.

The reviewer raises a valid point that false negative (FN) interactions (interactions which are real but based on our gold standard database are called as false positives) may account for some of low overlap between interactomes. As these interactions can't be distinguished with our approach we cannot assess their contribution to low overlap between interactomes.

9.) I would have expected a good coverage of the mitochondrial respiratory chain complexes in the dataset. However, looking at a few selected examples, the coverage does not seem to be good. More importantly, sometimes proteins that are part of different complexes are reported to interact. The only interaction partner in Table S6 of P00156 (Cytochrome B, a central protein in complex III) is P56181-2, which is part of complex I. COXII (P00403) only interacts with COX6C (P09669) and none of the other members of complex iv.

As PCP does not assess binary interactions but correlations between proteins it is common for indirect interactions, such as those from protein within the same supramolecular complex yet separate subunits, to be assigned. The mitochondrial respiratory chain complex is a good example of this as individual subunits can be observed in multiple associations, as shown in Figure 1C. In these cases, non-direct interactions can be assigned but it is important to note these associations are real. Within our analysis we have allowed for this by utilizing supramolecular information from CORUM which allows these associations to be correctly assigned as true positives. Within the example highlighted by reviewer of P00156 interacting with P56181-2 this interaction is only observed at a BN-PAGE position of 11.9 corresponding to a size of ~1.5mDa. This mass is consistent with a supramolecular complex of the intact mitochondrial respiratory chain complex I and III, an association previously reported by Schafer *et al* JBC 2006. The apparent absence of other known/expected interactions is likely a reflection of our conservative, yet potentially too stringent, approach (see 8).

For clarity and to highlight that indirect interactions can be assigned we have included the following statement within the results section “This approach, unlike techniques which

assess only direct interactions such as yeast two-hybrid, enables both direct and indirect interactions between proteins found within the same supramolecular complex to be identified.”

10.) I don't fully understand the logic behind Table S6. The first column header is "unique interactions" but actually several of these occur multiple times. For example, "P00403_P09669" is listed three times, all detected in both replicates. I don't understand why this is. What causes this redundancy?

The cause of this redundancy is that the same interactions can be observed at multiple positions within a SEC/BN-PAGE separation, as the same two proteins could be part of different complexes. This spatial information is a unique feature of PCP and provides useful insight into understanding the observed interactome. Within the tables EV6, 9, 10, 16 and 17 we provided both interaction information (labelled Unique interactions) and the spatial information (labelled Center A and Center B corresponding to the determined center of the Gaussian feature of each protein) for each interaction determined. Furthermore, within the manuscript we highlight the utility of these unique features of PCP and describe how they can be used to understand protein associations with example of the proteins O00483, P10606 and P09669 of Complex IV in Figure 1C and D.

11.) The first paragraph of the introduction is very general with little relevance to the investigated question (that is, changes in the interactome during apoptosis). I think it should be shortened.

We thank the reviewer for their suggestion and have modified the first paragraph to be more focused.

12.) Page 3, line 20, should be "differ in the role OF mitochondria"

We thank the reviewer for highlighting this error, it has been corrected.

13.) Fig. S3, legend: "Gel slices are generated and in- gel digestion preformed.", should be performed

We thank the reviewer for highlighting this error, it has been corrected.

14.) It would be very helpful and increase readability to number the main text figures.

The main figures have now been labeled.

15.) Fig. 1 A: "Gaussians features" should be "Gaussian features"

We thank the reviewer for highlighting this error, it has been corrected.

16.) Page 9: "This could occur in one of two ways: either proteolysis drove the rearrangement of the complexes or rearrangement has led to their subsequent proteolysis". There is of course also a third possibility: Proteolysis and complex rearrangement could be two rather independent processes that are both induced by apoptosis. The significant overlap with Degradase might have other reasons.

The reviewer is correct and this possibility has been added to the discussion. This sentence within the manuscript now reads "This could occur in one of three ways: proteolysis drove the rearrangement of the complexes, the rearrangement has led to their subsequent

proteolysis or proteolysis and complex rearrangement are two independent processes induced by apoptosis.”

Reference:

Babu M, Vlasblom J, Pu S, Guo X, Graham C, Bean BD, Burston HE, Vizeacoumar FJ, Snider J, Phanse S, Fong V, Tam YY, Davey M, Hnatshak O, Bajaj N, Chandran S, Punna T, Christopolous C, Wong V, Yu A et al (2012) Interaction landscape of membrane-protein complexes in *Saccharomyces cerevisiae*. *Nature* **489**: 585-589

Dix MM, Simon GM, Wang C, Okerberg E, Patricelli MP, Cravatt BF (2012) Functional interplay between caspase cleavage and phosphorylation sculpts the apoptotic proteome. *Cell* **150**: 426-440

Drew D, Newstead S, Sonoda Y, Kim H, von Heijne G, Iwata S (2008) GFP-based optimization scheme for the overexpression and purification of eukaryotic membrane proteins in *Saccharomyces cerevisiae*. *Nat Protoc* **3**: 784-798

Kristensen AR, Gsponer J, Foster LJ (2012) A high-throughput approach for measuring temporal changes in the interactome. *Nat Methods*

Li H, Zhu H, Xu CJ, Yuan J (1998) Cleavage of BID by caspase 8 mediates the mitochondrial damage in the Fas pathway of apoptosis. *Cell* **94**: 491-501

Sakahira H, Enari M, Nagata S (1998) Cleavage of CAD inhibitor in CAD activation and DNA degradation during apoptosis. *Nature* **391**: 96-99

Scott NE, Brown LM, Kristensen AR, Foster LJ (2015) Development of a computational framework for the analysis of protein correlation profiling and spatial proteomics experiments. *J Proteomics* **118**: 112-129

Wittig I, Braun HP, Schagger H (2006) Blue native PAGE. *Nat Protoc* **1**: 418-428

Zhu W, Leber B, Andrews DW (2001) Cytoplasmic O-glycosylation prevents cell surface transport of E-cadherin during apoptosis. *EMBO J* **21**: 5999-6007

Thank you for submitting your revised study. We have now heard back from reviewer #3 who was asked to evaluate your manuscript. As you will see below, the reviewer thinks that all issues have been satisfactorily addressed and supports publication of the study.

Before we formally accept the study for publication, we would like to ask you to address some remaining editorial issues.

REFeree REPORT

Reviewer #3:

The authors appropriately addressed the points I raised. Although this is still a rather descriptive story, I think the manuscript is in principle acceptable for publication.

MOLECULAR SYSTEMS BIOLOGY

Corresponding Author Name: Leonard Foster

Manuscript Number: MSB-16-7067R